# The *Firre* locus produces a *trans*-acting RNA molecule that functions in hematopoiesis

Jordan P. Lewandowski[1], James C. Lee [1,2], Taeyoung Hwang[1,3], Hongjae Sunwoo [4], Jill M. Goldstein[1,5], Abigail F. Groff[1,6], Nydia P. Chang[1], William Mallard [1], Adam Williams[7], Jorge Henao-Meija[8], Richard A. Flavell [9], Jeannie T. Lee [4,10,11], Chiara Gerhardinger[1], Amy J. Wagers[1,5,12] & John L. Rinn [1,3]*

RNA has been classically known to play central roles in biology, including maintaining telomeres, protein synthesis, and in sex chromosome compensation. While thousands of long noncoding RNAs (lncRNAs) have been identified, attributing RNA-based roles to lncRNA loci requires assessing whether phenotype(s) could be due to DNA regulatory elements, transcription, or the lncRNA. Here, we use the conserved X chromosome lncRNA locus *Firre*, as a model to discriminate between DNA- and RNA-mediated effects in vivo. We demonstrate that (i) *Firre* mutant mice have cell-specific hematopoietic phenotypes, and (ii) upon exposure to lipopolysaccharide, mice overexpressing *Firre* exhibit increased levels of proinflammatory cytokines and impaired survival. (iii) Deletion of *Firre* does not result in changes in local gene expression, but rather in changes on autosomes that can be rescued by expression of transgenic *Firre* RNA. Together, our results provide genetic evidence that the *Firre* locus produces a *trans*-acting lncRNA that has physiological roles in hematopoiesis.

[1] Department of Stem Cell and Regenerative Biology and Harvard Stem Cell Institute, Harvard University, Cambridge, MA, USA. [2] Department of Medicine, University of Cambridge School of Clinical Medicine, Addenbrooke's Hospital, Cambridge, UK. [3] BioFrontiers Institute, University of Colorado Boulder, Boulder, CO, USA. [4] Department of Molecular Biology, Massachusetts General Hospital, Boston, MA, USA. [5] Paul F. Glenn Center for the Biology of Aging, Harvard Medical School, 77 Louis Pasteur Avenue, Boston, MA, USA. [6] Department of Systems Biology, Harvard Medical School, Boston, MA, USA. [7] The Jackson Laboratory, JAX Genomic Medicine, Farmington, CT, USA. [8] Department of Pathology and Laboratory Medicine, Perelman School of Medicine University of Pennsylvania, Philadelphia, PA, USA. [9] Department of Immunobiology and Howard Hughes Medical Institute, Yale University, School of Medicine, New Haven, CT, USA. [10] Department of Genetics, Harvard Medical School, Boston, MA, USA. [11] Howard Hughes Medical Institute, Boston, MA, USA. [12] Joslin Diabetes Center, Boston, MA, USA. *email: john.rinn@colorado.edu

Transcription occurs at thousands of sites throughout the mammalian genome. Many of these sites are devoid of protein-coding genes and instead contain long noncoding RNAs (lncRNAs). While lncRNA loci have been implicated in a variety of biological functions, comparatively few lncRNA loci have been genetically defined to have RNA-based roles. Indeed, deletions of entire lncRNA loci have uncovered a number of in vivo phenotypes[1–5]; however, this approach alone is confounded because in addition to the lncRNA transcript, lncRNA loci can also exert function through DNA regulatory elements[6–8], the promoter region[9], as well as by the act of transcription[10,11]. Thus, attributing RNA-based role(s) to lncRNA loci requires testing whether other regulatory modes potentially present at the locus have molecular activity that could contribute to phenotypic effects[3,12,13].

In this study, we use the functional intergenic repeating RNA element, (Firre) locus as a model to discriminate between DNA- and RNA-mediated effects in vivo. We selected this locus for our study because it is syntenically conserved in a number of mammals, including human[14–17], and because studies have reported diverse biological and molecular roles. Early characterization of the FIRRE locus in human cell lines identified it as a region that interacts with the X-linked macrosatellite region, DXZ4, in a CTCF-dependent manner[18–21]. Further analyses of the Firre locus demonstrated that it produces a lncRNA that escapes X-inactivation[15,22–24], although it is predominately expressed from the active X chromosome[21,25]. Studies using cell culture models suggest that the Firre locus has biological roles in multiple processes, including adipogenesis[26], nuclear architecture[15,19,21], and in the regulation of gene expression programs[15,27]. In addition, there is some evidence for roles of the FIRRE locus in human development and disease[28–31]. Collectively, these studies demonstrate the diverse cellular and biological functions for the Firre locus. However, the biological roles of Firre as well as disentangling DNA- and RNA-mediated function(s) for the Firre locus have not been explored in vivo.

Using multiple genetic approaches, we describe an in vivo role for the Firre locus during hematopoiesis. We report that Firre mutant mice have cell-specific defects in hematopoietic populations. Deletion of Firre is accompanied by significant changes in gene expression in a hematopoietic progenitor cell type, which can be rescued by induction of Firre RNA from an autosomal transgene within the Firre knockout background. Mice overexpressing Firre have increased levels of pro-inflammatory cytokines and significantly impaired survival upon exposure to lipopolysaccharide (LPS). Finally, the Firre locus does not contain cis-acting RNA or DNA elements (including the promoter) that regulate neighboring gene expression on the X chromosome (nine biological contexts examined), suggesting that Firre does not function in cis. Collectively, our study provides evidence for a trans-acting RNA-based role for the Firre locus that, thus far, has physiological importance for hematopoiesis.

## Results

**The Firre locus produces an abundant lncRNA.** We first sought to investigate the gene expression properties for Firre RNA in vivo. To determine potential spatial and temporal aspects of Firre RNA expression during development, we performed in situ hybridization in wild-type (WT) mouse embryos (E8.0–E12.5). Notably, we detected Firre RNA in many embryonic tissues, including the forebrain, midbrain, pre-somitic mesoderm, lung, forelimb, hindlimb, liver, and heart (Fig. 1a). Since noncoding RNAs have been described to be generally expressed at lower levels compared with protein-coding genes[32–35], we determined the relative abundance of Firre RNA in vivo. We performed

RNA-seq on eight different WT embryonic tissues and plotted the expression of noncoding and coding transcripts. Consistent with previous reports[32–35], we observed that noncoding transcripts were generally less abundant than protein-coding transcripts (Fig. 1b). Despite most lncRNAs being expressed at low levels, we found that Firre, like Malat1 (refs[36–38]), is an abundant transcript (Fig. 1b). Next, since Firre is located on the X chromosome and escapes X-inactivation[15,22–24], we investigated whether Firre has different expression levels in male and female WT tissues. While levels of Firre RNA varied across embryonic tissue types, within individual tissues, male and female samples exhibited similar expression levels of Firre, despite escaping X-inactivation (Fig. 1c).

**Firre knockout and overexpression mice are viable and fertile.** To investigate the in vivo role of Firre and assess DNA- and RNA-mediated effects, we generated both Firre loss-of-function and Firre overexpression mice. To delete the Firre locus in vivo, we generated a mouse line containing a floxed allele (Firre^floxed) from a previously targeted mouse embryonic stem cell line[15] and mated to a CMV-Cre deleter mouse[39]. This produced a genomic deletion (81.8 kb) that removed the entire Firre gene body and promoter (henceforth called ΔFirre) (Fig. 1d). We confirmed the deletion of the Firre locus by genotyping (Supplementary Fig. 1) and examined Firre RNA expression. As expected, we did not detect Firre RNA in ΔFirre embryos by whole-mount in situ hybridization or by RNA-seq (Fig. 1a, d).

Since Firre is found on the X chromosome, we first sought to determine if deletion of the locus had an effect on the expected sex ratio of the progeny. Matings between ΔFirre mice produced viable progeny and had a normal frequency of male and female pups (Supplementary Table 1) that did not exhibit overt morphological, skeletal, or weight defects (Supplementary Fig. 2). Moreover, deletion of Firre did not impact expression levels of Xist RNA in embryonic tissues or perturb Xist RNA localization during random X chromosome inactivation (XCI) in mouse embryonic fibroblasts (MEFs) (Supplementary Fig. 3A–C).

Because the ΔFirre allele removes the entire gene body, this model does not allow us to distinguish between DNA- and RNA-mediated effects. Therefore, in order to be able to investigate the role of Firre RNA, we generated a doxycycline (dox)-inducible Firre overexpression mouse. This mouse model was engineered to contain a Firre cDNA downstream of a tet-responsive element (henceforth called tg(Firre)), and was mated to mice that constitutively express the reverse tetracycline transcriptional activator (rtTA) gene (combined alleles henceforth called Firre^OE) (Fig. 1e). This approach enabled systemic induction of Firre RNA in a temporally controllable manner by the administration of dox. Moreover, by combining the Firre^OE and ΔFirre alleles (henceforth called Firre^rescue), we could test whether Firre RNA expression alone is sufficient to rescue phenotypes arising in the ΔFirre mice, thereby distinguishing DNA- and RNA-based effects.

To confirm expression of transgenic Firre RNA, tg(Firre) females were mated with rtTA males and placed on a dox diet the day a copulatory plug was detected, and embryos were collected at E11.5 for analyses. Compared with sibling control embryos, we detected increased Firre RNA in Firre^OE embryos by whole-mount in situ hybridization (Fig. 1f) and by quantitative reverse transcription-PCR (qRT-PCR) (heart, 16-fold; forebrain, 26.6-fold; and forelimb, 11.5-fold) (Fig. 1g). Moreover, matings between tg(Firre) and rtTA mice fed a dox diet produce viable progeny that overexpress Firre at expected male and female frequencies (Supplementary Table 1).

Firre RNA has been reported to be largely enriched in the nucleus of mouse embryonic stem cells (mESCs)[15,40], neuronal

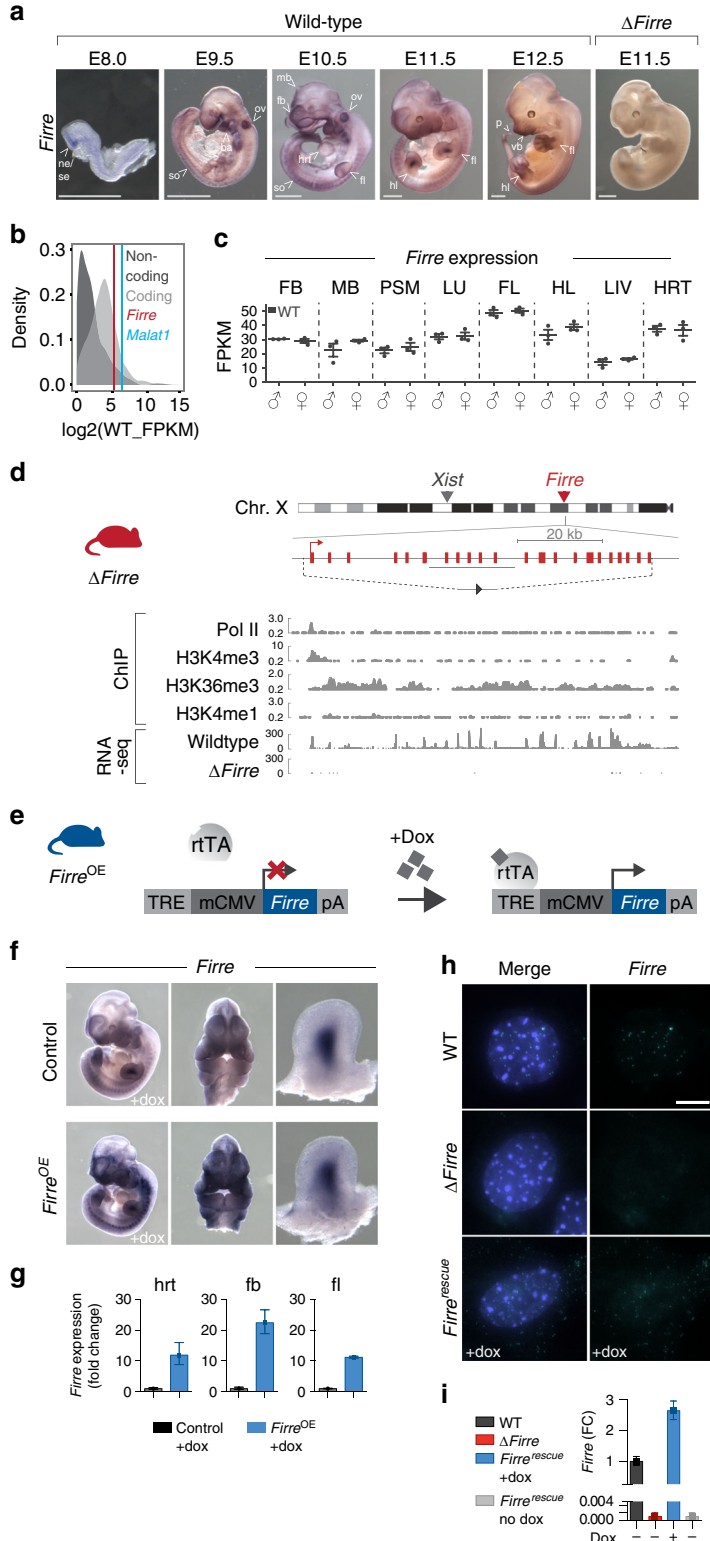

precursor cells[39], and HEK293 cells[17], and also has been reported in the cytoplasm of a human colon cell line[27]. Thus, we investigated the subcellular localization of *Firre* in the genetic models using RNA fluorescent in situ hybridization (RNA FISH). In contrast to Δ*Firre* MEFs, we detected pronounced localization of *Firre* RNA in the nucleus of WT MEFs (Fig. 1h). In dox-treated *Firre*rescue MEFs, which only produce *Firre* RNA from the transgene, we detected *Firre* RNA in both the nucleus and cytoplasm (Fig. 1h), which corresponded to approximately

a 2.7-fold increase in *Firre* RNA relative to WT (Fig. 1i). Notably, the *Firre*rescue transgenic model showed both nuclear and cytoplasmic localization of *Firre*, suggesting a threshold level control for nuclear localized *Firre*.

**Tissue-specific gene dysregulation in Δ*Firre* tissues.** Given the broad expression profile of *Firre* RNA (Fig. 1a), we took an initial unbiased approach to explore the potential biological roles for the

**Fig. 1** Mouse models to interrogate the in vivo function of *Firre*. **a** Whole-mount in situ hybridization for *Firre* RNA in WT mouse embryos at E8.0 ($n = 4$), E9.5 ($n = 4$), E10.5 ($n = 5$), E11.5 ($n = 6$), E12.5 ($n = 4$), and Δ*Firre* E11.5 embryos ($n = 3$). Scale bar equals 1 mm. **b** Distribution of transcript abundances for protein-coding (light gray) and noncoding (dark gray) genes in WT E11.5 heart tissue (representative tissue shown from seven additional tissues). Vertical lines indicate *Firre* (red) and *Malat1* (blue). **c** Expression of *Firre* shown as fragments per kilobase of transcript per million mapped reads (FPKM) from RNA-seq in E11.5 WT male ($n = 3$) and female ($n = 3$) forebrain (FB), midbrain (MB), pre-somitic mesoderm (PSM), lung (LU), forelimb (FL), hindlimb (HL), liver (LIV), and heart (HRT). The data shown as mean ± standard error of the mean (SEM). **d** *Firre* knockout mouse (red). Schematic of mouse X chromosome ideogram showing the *Firre* locus relative to *Xist*. UCSC genome browser diagram of the *Firre* locus (shown in opposite orientation). Dashed lines indicate the genomic region that is deleted in Δ*Firre* mice; single loxP scar upon deletion (gray triangle). Histone modifications and transcription factor binding sites in mouse embryonic stem cells (mESC-Bruce4, ENCODE/LICR, mm9). RNA-seq tracks for the *Firre* locus in WT and Δ*Firre* E11.5 forelimbs. **e** Schematic of doxycycline (dox)-inducible *Firre* overexpression mouse (*Firre*^OE^). Tet-responsive element (TRE), minimal CMV promoter (mCMV), reverse tetracycline transcriptional activator (rtTA), and β-globin polyA terminator (pA). **f** In situ hybridization for *Firre* at E11.5 in control (WT or tg(*Firre*) +dox) ($n = 4$) and *Firre*^OE^ +dox ($n = 3$) embryos. **g** qRT-PCR for *Firre* expression shown as fold change (FC) in dox-treated E11.5 control and *Firre*^OE^ hrt, fb, and fl. **h** RNA FISH for *Firre* in male WT, Δ*Firre*, and *Firre*^rescue^ MEFs. DAPI (blue) marks the nucleus and *Firre* RNA is shown in green. Scale bar equals 10 μm. **i** qRT-PCR for *Firre* expression shown as FC in male WT, Δ*Firre*, *Firre*^rescue^ +dox, and *Firre*^rescue^ no dox MEFs. Expression normalized to β-actin in the control or WT sample. The data plotted as mean ± CI at 98%. Source data are provided in the Source Data file

*Firre* locus, and performed poly(A)+ RNA-seq on eight E11.5 tissues from WT and Δ*Firre* embryos (forebrain, midbrain, heart, lung, liver, forelimb, hindlimb, and pre-somitic mesoderm). As expected, *Firre* expression was not detected in any of the Δ*Firre* tissues (Supplementary Data 1–8). Deletion of *Firre* was accompanied by significant changes in gene expression in all tissues examined (>1 FPKM, FDR<0.05) (Fig. 2a, b; Supplementary Data 1–8).

Across these eight tissues, we identified a total of 3910 significantly differentially expressed genes, of which 271 genes were differentially expressed in two or more tissues (Supplementary Data 1–8). Interestingly, gene ontology (GO) analysis of the commonly dysregulated genes showed that deletion of the *Firre* locus affected genes involved in hemoglobin regulation and general blood developmental processes (Fig. 2c). We therefore analyzed publicly available mouse RNA-seq data sets, and found that *Firre* is expressed across many blood cell types and note that expression is found highest in hematopoietic stem cells (HSCs)[41] and then decreases in conjunction with hematopoietic differentiation[42] (Fig. 2d). Based on this information, we narrowed our investigation to evaluate potential roles for *Firre* in the blood system, and leveraged the genetic mouse models to test DNA- and RNA-mediated effects.

**LPS potentiates the innate immune response in *Firre*^OE^ mice.** *Firre* is expressed in many innate immune cell types (Fig. 2d). Cell-based approaches have shown that *Firre* can be transcriptionally upregulated upon exposure to lipopolysaccharide (LPS) and can modulate the levels of inflammatory genes in a human colorectal adenocarcinoma cell line (SW480)[27], mouse macrophage cell line (RAW264.7)[27], as well as in an injury model using cultured primary rat microglial cells[30]. Thus, we hypothesized that dysregulation of *Firre* might alter the inflammatory response in vivo and reasoned that investigating the inflammatory response in *Firre* loss- and gain-of-function mice could provide insight into the DNA- and RNA-mediated effects of *Firre*. To test this, we employed a commonly used endotoxic shock model by administering LPS intraperitoneally to cohorts of WT, Δ*Firre*, *Firre*^OE^ no dox, and dox-fed *Firre*^OE^ mice in order to stimulate signaling pathways that regulate inflammatory mediators[43] (Fig. 2e).

We administered two different LPS preparations, one which broadly stimulates the pattern recognition receptors toll-like receptors (TLR) 2, 4, and nitric oxide synthase, and an ultrapure LPS preparation that specifically stimulates TLR4 (refs[44–46]). At 5 h post LPS injection, we measured serum cytokine levels. Notably, we observed that *Firre*^OE^ dox-fed mice administered broad-acting LPS had significantly higher levels of inflammatory cytokines, including TNFα, IL12-p40, and MIP-2 compared with WT (Fig. 2f). In contrast, we did not observe a significant difference for these cytokines in LPS-treated Δ*Firre* mice (Fig. 2f). Consistent with the increased cytokine response using broad-acting LPS, dox-fed *Firre*^OE^ mice administered TLR4-specific-acting LPS also had significantly higher levels of TNFα, IL12-p40, and MIP-2 compared with WT (Fig. 2g), albeit at lower serum concentrations compared with the broad-acting LPS (Fig. 2f, g). In addition, we confirmed that overexpressing *Firre* RNA alone (without LPS) does not result in increased serum levels of TNFα, IL12-p40, and MIP-2 (Supplementary Fig. 4).

Because increased levels of TNFα is a hallmark of endotoxic shock[47–49], we next tested whether the levels of *Firre* RNA had an impact on survival following LPS treatment. We administered 5 mg/kg of TLR4-specific-acting LPS to WT ($n = 30$), Δ*Firre* ($n = 18$), *Firre*^OE^ no dox ($n = 13$), and *Firre*^OE^ dox-fed ($n = 17$) mice, as well as a saline control group and monitored for 6 days. At this dose, across two independent cohorts, dox-treated *Firre*^OE^ mice showed a significantly higher susceptibility to LPS compared with WT mice ($P<0.0001$, Mantel–Cox) and uninduced *Firre*^OE^ animals ($P = 0.0063$, Mantel–Cox) (Fig. 2h). Whereas Δ*Firre* mice did not show a significant difference in the level of susceptibility to LPS ($P = 0.1967$, Mantel–Cox test) (Fig. 2h). Collectively, these results indicate that endogenous *Firre* does not appear to be necessary for a physiological inflammatory response to LPS, but that ectopic levels of *Firre* RNA can modulate this inflammatory response in vivo independent of genomic context, consistent with an RNA-based role for *Firre*.

**Firre mice have cell-specific hematopoietic phenotypes.** Having observed an effect of *Firre* in regulating gene expression and accentuating the inflammatory response, we further investigated the role of *Firre* in hematopoiesis (Fig. 3a). We first examined cell populations in the peripheral blood in Δ*Firre* mice and observed a modest but significant reduction in the frequencies of CD4 and CD8 T cells, whereas the frequencies of B and NK cells were unaffected compared with WT (Fig. 3b; Supplementary Fig. 5A). To investigate the cause of this reduction, we examined the thymus (to assess for a defect in T-cell development) and the bone marrow (to assess for a defect in hematopoietic progenitor cells). There was no block in thymic development in Δ*Firre* mice, as normal frequencies of cells were observed at each developmental stage (Supplementary Fig. 5B, upper panels). However, we noticed that the absolute number of cells was generally lower in Δ*Firre* mice at every developmental stage, suggestive of a pre-thymic defect in progenitor development (Supplementary Fig. 5B, lower panels). Consistent with this, in the bone marrow compartment, we observed a significant reduction in both the

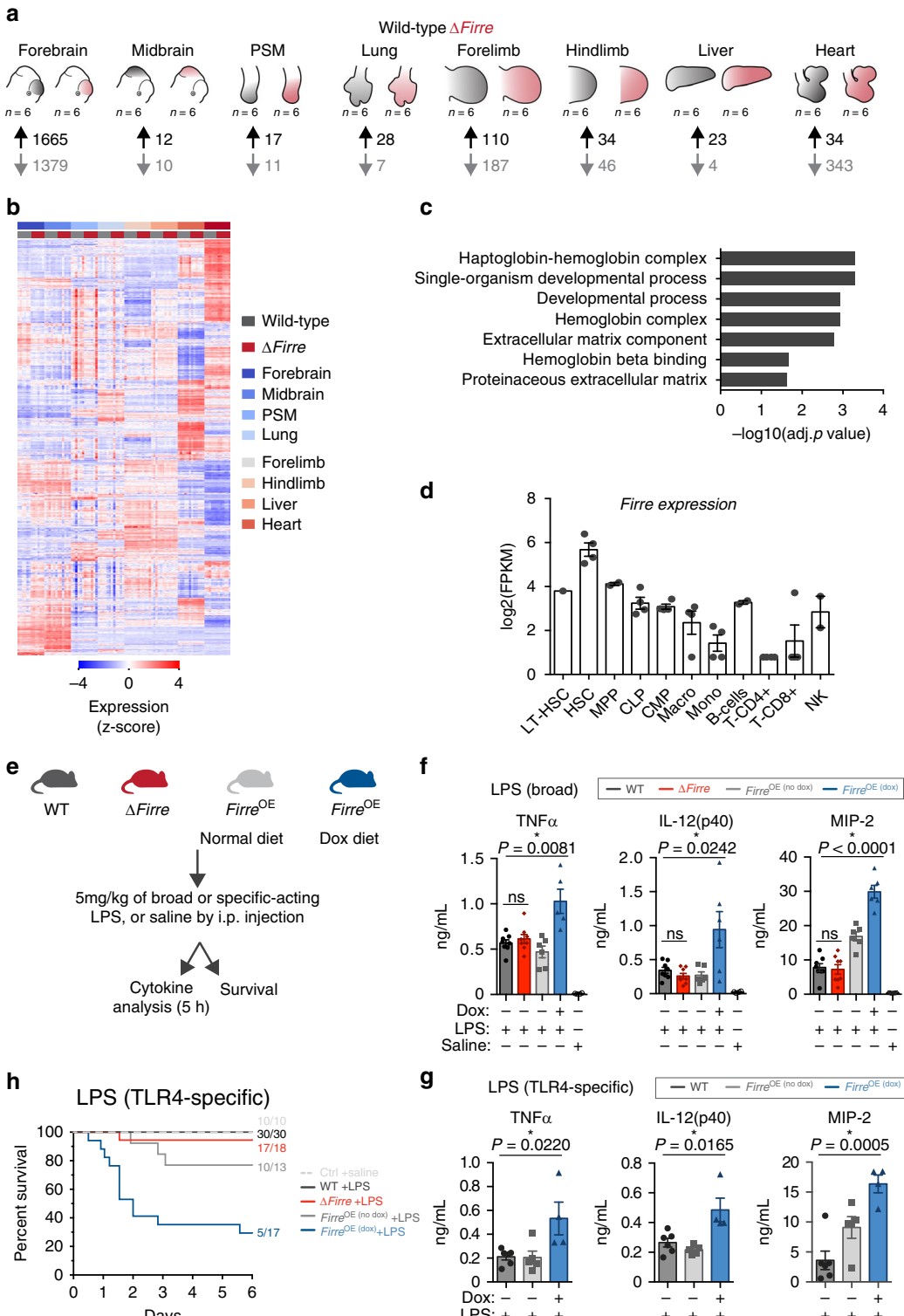

**Fig. 2** Modulation of *Firre* impacts genes with roles in the blood. **a** Schematized E11.5 tissues used for RNA-seq. WT (*n* = 6) shown in black, and Δ*Firre* (*n* = 6) shown in red. The number of differentially expressed genes shown below each tissue. **b** Heatmap of replicate embryonic tissues. **c** GO analysis for genes found dysregulated in four or more tissues. **d** *Firre* expression across multiple mouse blood cell lineages (RNA-seq data from bloodspot.eu, GSE60101). **e** Experimental approach for cytokine and survival experiments. **f** Cytokine measurements in serum at 5 h post intraperitoneal (i.p.) injection of 5 mg/kg LPS (broad acting) in WT (*n* = 8), Δ*Firre* (*n* = 8), *Firre*^OE control diet (*n* = 6), *Firre*^OE dox diet (*n* = 5–6), and saline-injected WT (*n* = 2) and Δ*Firre* (*n* = 2) (two independent experiments shown). **g** Cytokine measurements in serum at 5 h post i.p. injection of 5 mg/kg LPS (specific acting) in WT (*n* = 6), *Firre*^OE control diet (*n* = 5), *Firre*^OE dox diet (*n* = 4). Cytokine data are plotted as mean ± SEM, and significance determined by an unpaired two-tailed *t* test. **h** 6-day survival plot of mice injected with 5 mg/kg LPS (specific acting) in WT (*n* = 30), Δ*Firre* (*n* = 18), *Firre*^OE control diet (*n* = 13), and *Firre*^OE dox diet (*n* = 17) or saline control group (*n* = 10) consisting of WT, Δ*Firre*, and *Firre*^OE mice over two independent experiments. Statistical difference determined by a by a Mantel–Cox test with *P* < 0.05 deemed significant. Source data are provided in the Source Data file

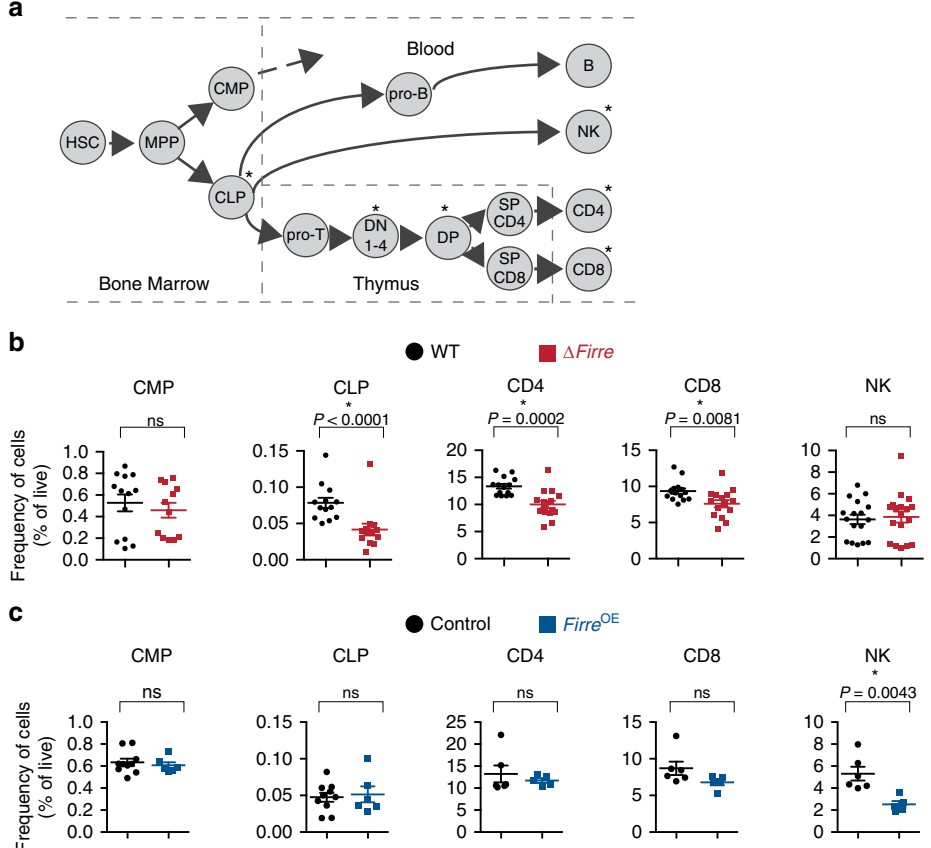

**Fig. 3** Δ*Firre* and *Firre*[OE] mice have cell-specific hematopoietic phenotypes. **a** Schematic of hematopoiesis. **b** Frequencies of common myeloid progenitors (CMP) and common lymphoid progenitors (CLP) in the bone marrow shown from WT (*n* = 13, black circles) and Δ*Firre* (*n* = 12-13, red squares) mice. Two representative experiments combined (three independent experiments). Frequencies of CD4 and CD8 cells from the peripheral blood from WT (*n* = 14) and Δ*Firre* (*n* = 15) mice. Frequency of NK cells from the peripheral blood from WT (*n* = 17) and Δ*Firre* (*n* = 18) mice. Three representative experiments combined (seven independent experiments). **c** Frequencies of CMPs and CLPs from the bone marrow from control (tg(*Firre*) or WT or rtTA with dox) (*n* = 10, black circle) and *Firre*[OE] +dox (*n* = 6, blue square) mice. One representative experiment is shown (two independent experiments). Frequencies of CD4, CD8, and NK cells from the peripheral blood from control (WT or tg(*Firre*) or rtTA with dox) (*n* = 6, black circle) and *Firre*[OE] +dox (*n* = 5, blue square) mice. One representative experiment is shown (three independent experiments). Cell frequencies determined by flow cytometry analysis. The data are plotted as percent (%) of live cells showing the mean ± SEM, and statistical significance determined by a two-tailed Mann–Whitney U test. Source data are provided in the Source Data file

frequency and number of the common lymphoid progenitors (CLPs) (lineage(lin)−Sca1loc-KitloIL7Rα+), a hematopoietic progenitor cell type, in Δ*Firre* mice (Fig. 3b; Supplementary Fig. 5C).

To assess whether the observed defect in hematopoiesis could be due to a progenitor-intrinsic effect of *Firre* deficiency, we performed a competitive chimera transplant assay using an HSC-enriched population. We isolated an HSC-enriched population (lin−Sca1+c-Kit+CD34+/−CD135−) from WT (CD45.2), Δ*Firre* (CD45.2), and congenic WT (CD45.1), and then separately mixed WT (CD45.2) or Δ*Firre* (CD45.2) cells at an equal ratio with congenic WT (CD45.1). We then transplanted this cell mixture into lethally irradiated CD45.1 recipient mice (Supplementary Fig. 6A, B) and assessed the long-term reconstitution ability of WT and Δ*Firre* HSCs to repopulate blood cell lineages in vivo. We observed that Δ*Firre*/CD45.2-donors had significant reductions in the frequencies of CD4 and CD8 T cells, B cells, and NK cells in the peripheral blood of recipient mice compared with WT/CD45.2-donors (*P* = 0.0028, *P* = 0.0051, *P* = 0.0114, *P* = 0.0068, respectively, two-tailed Mann–Whitney U), indicating that Δ*Firre*-donors were markedly outcompeted at repopulating the blood (Supplementary Fig. 6C). Together, these data are consistent with a progenitor-intrinsic role for *Firre* in hematopoiesis.

In contrast to the Δ*Firre* model, mice overexpressing *Firre* RNA in the WT background (*Firre*[OE]) had normal frequencies of CD4, CD8, and B cells, but had a significant reduction in the frequency of NK cells in the peripheral blood compared with control mice (Fig. 3c; Supplementary Fig. 5D). A decrease in the frequency of NK cells in dox-fed *Firre*[rescue] mice, where only *Firre* RNA from the transgene is expressed, was also observed (Supplementary Fig. 7). In the bone marrow of dox-fed *Firre*[OE] mice, we did not observe significant changes in the frequencies of HSC, multipotent progenitor (MPP), common myeloid progenitor (CMP), or CLPs compared with control samples (Fig. 3c; Supplementary Fig. 5E). Taken together, these results identify cell type-specific defects during hematopoiesis, whereby alterations of *Firre* impact the ratios and numbers of particular blood cells produced during hematopoiesis.

**_Firre_ RNA has a _trans_-acting role in vivo.** Next, we wanted to further investigate the DNA- and RNA-mediated effects of the *Firre* locus using a cell type that was dysregulated in the Δ*Firre* immunophenotyping analysis. We selected to use the CLP as a model, because this was the earliest hematopoietic defect identified and because *Firre* is highly expressed in this progenitor cell

type. Therefore, further investigation could provide insight into the physiological effects of modulating *Firre* in a progenitor cell population. Because the Δ*Firre* mouse contains a deletion that removes all potential DNA regulatory elements, the lncRNA, and the promoter (thus removing the act of transcription), this mouse model does not allow us to distinguish between DNA- and RNA-mediated effects.

To directly test whether the decrease in the frequencies of CLPs observed in the Δ*Firre* mice is mediated by an RNA-based mechanism, we reasoned that overexpressing *Firre* RNA in the Δ*Firre* background would enable to test for an RNA-mediated role. To this end, we generated multiple cohorts of mice that contained the *Firre*^OE alleles in the Δ*Firre* background (*Firre*^rescue) and induced transgenic *Firre* RNA expression by placing mice on a dox-diet. From multiple cohorts of WT, Δ*Firre*, and dox-fed *Firre*^rescue mice, we assessed CLP frequency by flow cytometry in total bone marrow and lineage-depleted bone marrow (to enrich for hematopoietic non-lineage committed cells).

Consistent with the previous data (Fig. 3b), we also observed a significant decrease in the frequency of CLPs in total bone marrow from Δ*Firre* mice ($n = 17$, mean CLP frequency = 0.0532) compared with WT mice ($n = 16$, mean CLP frequency = 0.0829) (Fig. 4b) and in lineage-depleted bone marrow from Δ*Firre* mice ($n = 9$, mean CLP frequency = 0.3011) compared with WT ($n = 9$, mean CLP frequency = 0.2167) (Supplementary Fig. 8A, B). In contrast, dox-fed *Firre*^rescue mice which only expressed transgenic *Firre* RNA, we observed that the frequency of CLPs was significantly increased ($n = 15$, mean CLP frequency = 0.0810) compared with Δ*Firre* mice, and restored to approximately that of WT in total bone marrow (Fig. 4b). Consistent with this data, in lineage-depleted bone marrow, we also observed a significant increase in the frequency of CLPs in dox-fed *Firre*^rescue mice ($n = 11$, mean CLP frequency = 0.2809) compared with Δ*Firre* mice (Supplementary Fig. 8). Thus, induction of transgenic *Firre* RNA alone is sufficient to rescue the reduction in frequency of CLPs observed in Δ*Firre* bone marrow. These data suggest that *Firre* RNA, rather than DNA, exerts a biological function during hematopoiesis.

**Transgenic *Firre* RNA restores gene-expression programs in vivo.** To gain further insight into the molecular roles of *Firre* in the CLPs, we took a gene expression approach, because alterations of the *Firre* locus and RNA have previously been shown to impact gene expression[15,21,50]. Moreover, we reasoned that we could test if changes in gene expression in the loss-of-function model could be rescued by expressing only transgenic *Firre* RNA. To this end, we isolated CLPs by fluorescence-activated cell sorting (FACS) from the bone marrow of age- and sex-matched WT, Δ*Firre*, and dox-fed *Firre*^rescue mice, and performed poly(A) + RNA-seq. As expected, *Firre* RNA was not detected in the Δ*Firre* samples, and expression of transgenic *Firre* RNA in the *Firre*^rescue samples was detected at levels above WT (Fig. 4c). Differential gene expression analysis between WT and Δ*Firre* CLPs identified 89 significantly differentially expressed genes (FDR<0.1) (Fig. 4d; Supplementary Data 9). GO analysis of the differentially expressed genes showed that deletion of *Firre* in CLPs affected genes involved in lymphocyte activation, cell adhesion, and B-cell activation (Fig. 4e).

Next, we determined if induction of *Firre* RNA in the *Firre*^rescue model could rescue expression of the 89 significantly dysregulated genes found in Δ*Firre* CLPs. We compared the CLP RNA-seq from Δ*Firre* and *Firre*^rescue mice and identified 4656 genes with significant changes in gene expression (FDR<0.1) (Supplementary Data 10). Notably, 78 of the 89 genes that were significantly differentially expressed in Δ*Firre* CLPs were found to be significantly and reciprocally regulated in *Firre*^rescue CLPs ($P = 2.2e-16$, Fisher exact test) (Fig. 4d). For example, *Ccnd3*, *Lyl1*, and *Ctbp1* are significantly downregulated in Δ*Firre* CLPs, but are found significantly upregulated in *Firre*^rescue CLPs to (Fig. 4f). Further, genes such as *Maoa*, *Fam46c*, and *Icos* were found significantly upregulated in Δ*Firre* CLPs, but their expression was significantly reduced in *Firre*^rescue CLPs (Fig. 4f). We also noted that several immunoglobin heavy and light-chain variable region genes were reciprocally regulated in our analyses (Fig. 4d). Taken together, these data suggest that ectopic expression of *Firre* is sufficient to restore a gene expression program in an RNA-based manner in vivo.

**The *Firre* locus does not function in *cis*.** Many lncRNA loci exert function to control the expression of neighboring genes, a biological function called *cis*-regulation[51]. This occurs through a variety of mechanisms including, *cis*-acting DNA-regulatory elements, the promoter region, the act of transcription, and the lncRNA[9,10,52–54]. The Δ*Firre* mouse model enables to test for potential *cis*-regulatory roles for *Firre* on the X chromosome, because the knockout removes the entire *Firre* locus and promoter region (Fig. 1d). To investigate local (*cis*) effects on local gene expression, we generated a 2 Mb windows centered on the *Firre* locus and examined whether the neighboring genes were significantly dysregulated across nine biological contexts.

Differential gene expression analysis for WT and Δ*Firre* CLPs showed that of the 12 genes within a 2 Mb window (excluding *Firre*), none were differentially expressed (Fig. 4g). Consistent with this finding, we did not observe significant changes in local gene expression (2 Mb windows centered on the *Firre* locus) in seven of the eight embryonic tissues (Fig. 4g; Supplementary Fig. 9A–F). Indeed, we observed one instance of differential expression in one embryonic tissue (*Hs6st2* was slightly but significantly downregulated in the embryonic forebrain, −0.38 log2 fold change, FDR<0.05) (Supplementary Fig. 9A–F). These data demonstrate that the *Firre* locus does not exert a local effect on gene expression in vivo, and suggest that the *Firre* lncRNA regulates gene expression in a *trans*-based manner. Collectively, our study investigates the roles of DNA and RNA at the *Firre* locus in vivo and genetically defines that the *Firre* locus produces as a *trans*-acting lncRNA molecule in vivo.

## Discussion

Classic models used to study noncoding RNAs—ribosomal RNAs, small nucleolar RNAs, tRNAs, and the telomerase RNA component (TERC)—have demonstrated that RNAs serve important cellular functions. This core of possible RNA biology has been greatly expanded by studies that have identified tens of thousands of lncRNAs[32,33,55]. Indeed, subsequent molecular and genetic interrogation of lncRNA loci have identified diverse molecular roles and biological phenotypes. However, lncRNA loci can potentially function through multiple molecular modes: DNA-regulatory elements (including the promoter), the act of transcription itself, and the lncRNA gene product. Therefore, attributing an RNA-based role to a lncRNA locus requires the development of multiple genetic models to determine the activities and contributions of potential regulatory modalities[12,13,52].

In this study, we developed three genetic models in mice for the syntenically conserved lncRNA *Firre*: loss-of-function, overexpression, and rescue. Notably, we report that deletion of the *Firre* locus does not impact survival in mice, or despite escaping XCI, skew the sex ratio of progeny. We leveraged the genetic models to discriminate between DNA- and RNA-mediated effects in vivo. We determined that modulating *Firre* directs cell-specific

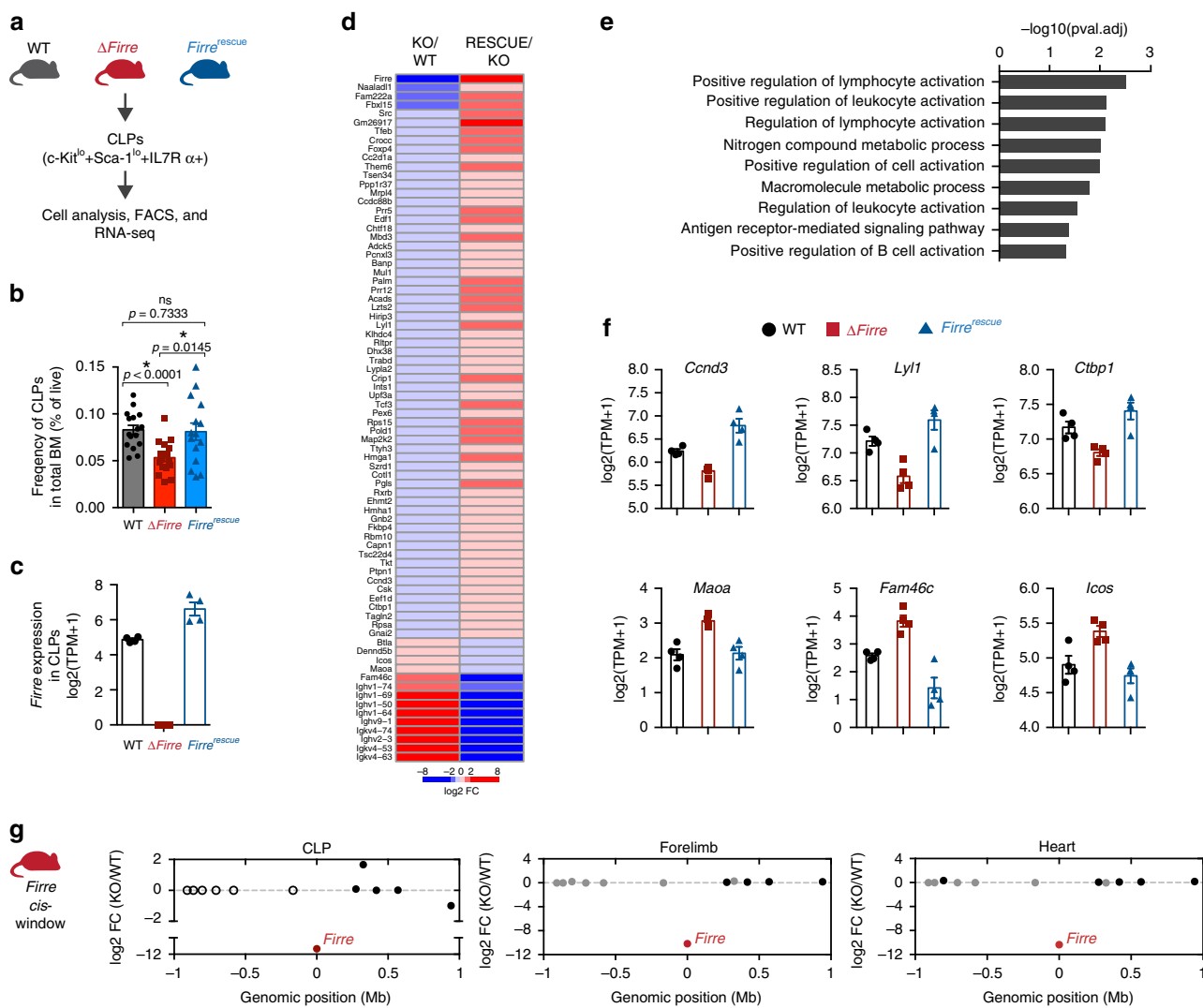

**Fig. 4** Transgenic *Firre* RNA rescues physiological and molecular defects in CLPs. **a** Schematic of experimental approach. **b** Frequencies of CLPs as determined by flow cytometry shown as percent of live cells in total bone marrow from 3 to 7 months old WT (*n* = 16, mean age = 26 weeks), Δ*Firre* (*n* = 17, mean age = 23 weeks), and *Firre*^rescue^ dox diet (*n* = 15, mean age = 23 weeks) mice over three independent experiments. The data are shown as mean ± SEM, and statistical significance determined by a two-tailed Mann–Whitney U test. **c** *Firre* RNA expression in CLPs from WT (*n* = 4), Δ*Firre* (*n* = 4), and dox-treated *Firre*^rescue^ (*n* = 4) determined by RNA-seq. The data plotted as transcripts per million (TPM +1) showing the mean ± SEM. **d** Heatmap showing significantly differentially expressed genes in CLPs in Δ*Firre*/WT comparison and dox-treated *Firre*^rescue^/Δ*Firre* comparison. Color gradient is saturated outside of +/− log2 fold changes. **e** GO analysis for significantly dysregulated genes in Δ*Firre* CLPs. **f** Examples of genes that show significant reciprocal regulation in WT, Δ*Firre*, and dox-treated *Firre*^rescue^ CLPs. **g** *Firre* locus region (2 Mb) showing gene expression differences in log2 FC between Δ*Firre* and WT CLPs, mouse embryonic forelimb, and heart. *Firre* is shown in red, significantly dysregulated genes are shown in red, genes that are not significantly changed are shown in black, and genes that were not detected shown in white. Source data are provided in the Source Data file

defects during hematopoiesis, potentiates the innate immune response upon exposure to LPS, and can restore gene expression programs—all of which have an RNA-based functional modality. We also conclude that the *Firre* locus does not have a local *cis*-regulatory effect on gene expression in nine different biological contexts. Together, by using multiple genetic and molecular approaches, we identified that *Firre* produces a *trans*-acting lncRNA in a hematopoietic context. There are several important implications for these results.

First, our study indicates that *Firre* has *trans* RNA-based activity in vivo, and thus extends reports that have suggested RNA-based roles for *Firre* in cell culture models[25,27,40]. By using compound genetic approaches, we found that overexpression of *Firre* from a transgene in the *Firre*-deficient background was sufficient to rescue physiological and molecular phenotypes in Δ*Firre* CLPs in vivo. The rescue of genes found differentially

expressed in Δ*Firre* CLPs by a *Firre* transgene produced a highly significant result (*P* = 2.2e-16, Fisher exact test); however, we note that the widespread changes in gene expression observed in the CLPs from animals only expressing transgenic *Firre* RNA could also formally contribute to this effect. Yet, we speculate that early hematopoietic progenitor cells may represent a unique context to study the role(s) of *FIRRE*/*Firre*. In humans, *FIRRE* is expressed as both circular (circ-*FIRRE*) and linear forms in hematopoietic cells, and circ-*FIRRE* is abundant in all progenitor cell types, except for the CLPs[56]. More studies will be needed to determine how and if *Firre* RNA is interacting at distal loci as well as assess the functional differences between linear and circular isoforms of *Firre* in a progenitor cell context.

Second, we observed that overexpression of *Firre* RNA in an endotoxic shock model potentiated the innate immune response in vivo, and in turn, led to significant viability defects. Consistent

with our findings, independent studies have found that over-expression of *Firre* RNA in different cellular contexts can modulate genes in the innate immune response. For example, the levels of select inflammatory genes, including IL12p40, are increased upon overexpression of *Firre* RNA followed by LPS stimulation in human SW480 cells[27]. In addition, upon overexpression of *Firre* RNA in an injury model using cultured primary microglial cells, TNF-α levels are increased[30]. The different, yet synergistic approaches (cell-based, ex vivo, and in vivo) provide supporting evidence for an RNA-based role for *Firre*. In the overexpression approach in this study, we note that the phenotypes could be due to increased levels of *Firre* in a cell type where it is normally expressed and/or from de novo expression in a cellular context where *Firre* is not endogenously found. Yet, this strategy may perhaps mimic disease contexts where *FIRRE* is found overexpressed. We speculate that *Firre* could be important in setting functional thresholds for cells. For example, *FIRRE* is not only significantly increased in certain cancers, including, kidney renal cell carcinoma, lung squamous cell carcinoma, and colon adenocarcinoma[31], but high levels of *FIRRE* expression have been significantly associated with more aggressive disease and poor survival in patients with large B-cell lymphoma[29].

Finally, our study suggests that *Firre* does not have a *cis*-regulatory role on gene expression in vivo. Upon deletion of the *Firre* locus and its promoter region, we observed global changes in gene expression. Yet, we did not find changes in local gene expression (2 Mb window) at the *Firre* locus in most embryonic tissues and in CLPs. Thus, potential DNA regulatory elements, the lncRNA, the promoter, and the act of transcription appear to not have regulatory roles on neighboring gene expression in the nine biological contexts (eight embryonic and one cell type) analyzed in this study. Moreover, we observed that deletion of *Firre* in vivo does not perturb *Xist* RNA expression in eight embryonic tissues and does not affect random XCI in MEFs, which is consistent with a previous study using cell culture models[21]. Together, these data are notable because *cis*-acting mechanisms are speculated to be common feature at lncRNA loci[57]. While we did not find any evidence for *cis*-activity at the *Firre* locus in vivo, a previous study from our group found active DNA elements within the *Firre* locus using a cell-based enhancer reporter assay in 3T3 cells[14]. We speculate that these candidate DNA-regulatory elements are likely to regulate the *Firre* locus rather than neighboring genes, as we did not find evidence that neighboring genes were significantly dysregulated upon deletion of the *Firre* locus.

In summary, we have developed genetic models to test the DNA- and RNA-mediated effects of the syntenically conserved lncRNA locus, *Firre*, in vivo. Our findings provide evidence that the *Firre* lncRNA locus has a role in hematopoiesis that is mediated by a *trans*-acting RNA, and further highlights the biological importance of lncRNA-based machines in vivo. While this study defines a role for *Firre* in a hematopoietic context, it is important to note that deletion of *Firre* perturbed gene expression in a number of tissues, including the brain, where *FIRRE* has been implicated in a rare human disease[28]. Therefore, going forward it will be important to use genetic models to investigate the potential role(s) of *Firre* in a number of different biological and disease contexts.

## Methods

**Mouse care and ethics statement**. Mice used in this study were maintained in a pathogen-specific free facility that is under the supervision of Harvard University's Institutional Animal Care Committee. We have complied with the ethical regulations for animal testing and research in accordance with Harvard University's Institutional Animal Care and Use Committee.

Mice used in this study were housed at a density of 2–5 mice per cage containing: Anderson's Bed (The Andersons, Inc), Enviro-Dri (Shepherd Specialty Papers), compressed 2" × 2" cotton nestlet (Ancare), a red mouse hut (BioServ),

automatic watering of reverse osmosis deionized water was chlorinated at 2 ppm, and were fed a regular chow diet (Prolab IsoPro RMH 3000 5P75/76).

**Firre mouse strains and genotyping**. A previously targeted allele for *Firre* in JM8 embryonic stem cells (mESC) was generated by sequential targeting where a floxed-*neomycin*-floxed cassette was inserted at the 5′ end of the *Firre* locus between nucleotides 4790843 and 4790844 (mm9), and a floxed-*hygromycin*-floxed cassette was inserted at the 3′ end of the *Firre* locus between nucleotides 47990293 and 47990294 (mm9)[15]. mESCs containing both correctly targeted neomycin and hygromycin cassettes (*Firre^floxed*) were injected into blastocysts (Harvard Genome Modification Facility), and subsequent progeny were screened for the *Firre^floxed* allele (Jackson Laboratory, 029931). To generate a deletion of the *Firre* locus in all tissues, female *Firre^floxed* mice were mated to male B6.C-Tg(CMV-Cre)1Cgn/J mice[39] (CMV-Cre) (Jackson Laboratory, 006054). We screened the progeny for the ΔFirre allele by PCR genotyping (described below) for *Firre* wild-type, knockout (ΔFirre), neomycin, hygromycin, and cre alleles (Supplementary Fig. 1A, B). A female containing the ΔFirre and CMV-Cre alleles was subsequently mated to male C57BL/6J mice in order to remove CMV-Cre and to propagate the ΔFirre allele. Progeny were PCR genotyped for *Firre* wild-type, ΔFirre, and cre alleles. Progeny containing only the ΔFirre allele were backcrossed to C57BL/6J mice for three generations (N3) (Jackson Laboratory, 030038). Mice were inbred by intercrosses by heterozygous and homozygous breeding strategies.

To generate an inducible *Firre*-overexpressing allele (tg(*Firre*)) in mice, we cloned a mouse *Firre* cDNA[15] into a Tet-On vector (pTRE2), where the β-globin intron sequence was removed. The construct was verified by Sanger DNA sequencing. Next, we used EcoRI and NheI restriction enzymes to digest the cassette containing the tet-responsive element, CMV minimal promoter, *Firre* cDNA, and β-globin poly(A) terminator. The digested DNA fragment was purified according to Harvard Genome Modification Facility protocol, and the purified DNA cassette was injected into the pronucleus of C57BL/6J zygotes (Harvard Genome Modification Facility). Progeny were screened for the tg(*Firre*) cassette by PCR genotyping for the tg(*Firre*) allele, and male founder mice were identified and individually mated to female C57BL/6J mice (Jackson Laboratory, 000664). To overexpress tg(*Firre*) N2 and N3 generation, females were mated to male B6N.FVB (Cg)-Tg(CAG-rtTA3)4288Slowe/J (*rtTA*) mice (Jackson Laboratory, 016532) and at the plug date females, and the subsequent progeny were either placed on a normal chow diet or 625 mg/kg doxycycline-containing chow (Envigo, TD.01306) until the experimental end points. A colony of male *rtTA* mice were maintained by breeding to C57BL6/J females for up to four generations.

Genotyping for mice was performed on tissue biopsies that were mixed with 200 μL of digest buffer (50 mM KCl, 10 mM Tris–HCl pH 8.3, 2.5 mM MgCl₂, 0.1 mg/mL gelatin, 0.45% NP40, and 0.45% Tween-20 in ddH₂O with 10 μg/mL proteinase K) in a 1.5 -mL eppendorf tube, and were incubated for 6–18 h at 54 °C. Next, samples were placed on a heating block at 100 °C for 5 minutes (min) in order to heat inactivate proteinase K. Genotyping was performed by PCR using 2x PCR Super Master Mix (Biotool, B46019) and Quick-Load Taq 2X Master Mix (NEB, M0271L) with the cycling conditions 94 °C for 3 min, followed by 35 cycles of 94 °C for 20 s, 55 °C for 30 s, 72 °C for 45 s. Primers used for genotyping: *Firre* wild-type allele, F: GGAGGAGTGCTGCTTACTGG, R: TCTGTGAGCCACCTG AAATG; ΔFirre allele, F: TCACAATGGGCTGGGTATTCTC, R: CCTGGGTCC TCTATAAAAGCAACAG; *neomycin*, F: GACCACCAAGCGAAACATC, R: CT CGTCAAGAAGGCGATAGAA; hygromycin, F: CGGAAGTGCTTGACATTG GG, R: CGTCCATCACAGTTTGCCAGTG; Cre, F: TAATCCATATTGGCAGA ACG, R: ATCAATCGATGAGTTGCTTC; Sry, F: TTGTCTAGAGAGCATGGAG GGCCATGTCAA, R: CCACTCCTCTGTGACACTTTAGCCCTCCGA; tg(*Firre*) allele, F: TACCACTCCCTATCAGTGA, R: CGGCTTCATCTTCAGTCCTC; and the rtTA allele, F: AGTCACTTGTCACACAACG, R: CTCTTATGGAGATCCC TCGAC. Additional genotyping was performed by Transnetyx using real-time PCR.

**Cytokine analysis and in vivo endotoxin challenge**. To investigate the cytokine response in vivo, we used two different preparations of LPS from *Escherichia coli* (*E. coli*) O111:B4. (Sigma, L2630) and Ultrapure LPS, *E. coli* O111:B4 (InvivoGen, tlrl-3pelps) and dissolved in 0.9% saline solution (Teknova, S5825). We administered either 0.9% saline or 5 mg/kg broad-acting LPS (Sigma, L2630) by i.p. injection using 30-G insulin syringes (BD, 328411) to two female mice cohorts 8–10 weeks old (WT, ΔFirre, *Firre^OE* no dox, and dox fed *Firre^OE*). We also administered a different preparation of LPS at 5 mg/kg that is TLR4-specific (InvivoGen, tlrl-3pelps) by i.p. injection in mice 5–10 weeks old (WT, *Firre^OE* no dox, and dox fed *Firre^OE*). At 5 h post i.p. injection, mice were euthanized, and peripheral blood was collected by cardiac puncture and allowed to clot for 30 min at room temperature with gentle rotation. After clotting, samples were centrifuged at 1000×*g* for 10 min at 4 °C, and serum was collected. Cytokine analysis was performed on serum diluted twofold in PBS pH 7.4 (Eve Technologies, Chemokine Array 31-Plex). Measurements within the linear range of the assay are reported.

Endotoxin survival experiments were performed over two independent experiments using mice 9–16 weeks old: WT (mean age = 12.9 weeks), ΔFirre (mean age = 16 weeks), *Firre^OE* no dox (mean age = 13.6 weeks), and dox fed *Firre^OE* (mean age = 13.7 weeks). Saline control group consisting of WT, ΔFirre, and *Firre^OE* mice. In all, 0.9% saline or 5 mg/kg LPS (InvivoGen, tlrl-3pelps) was

prepared as described above and administered by i.p. injection, and mice were monitored for moribund survival over 6 days. Mice were housed at a density of 3–5 mice per cage containing: Anderson's Bed (The Andersons, Inc), Enviro-Dri (Shepherd Specialty Papers), compressed 2" × 2" cotton nestlet (Ancare), and a red mouse hut (BioServ). The following supportive care was provided during the duration of the experiments: hydrogel, a small cup containing powdered diet mixed with water, and a heating pad (5" × 8.6" × 6") was placed externally on the bottom of the cage (Kobayashi).

**Whole-mount in situ hybridization**. We generated a digoxigenin-labeled anti-sense riboprobe by in vitro transcription (Roche, 11277073910) against *Firre* from a 428 bp sequence (Supplementary Fig. 1C) that corresponds to the 5′ end of the *Firre* transcript. In situ hybridization was performed on a minimum of three embryos per stage and/or genotype. For whole-mount staining, we fixed embryos in 4% paraformaldehyde for 18 h at 4 °C, followed by three washes in 1× PBS for 10 min at room temperature. We then dehydrated the embryos for 5 min at room temperature in a series of graded methanol solutions (25%, 50%, 75%, methanol containing 0.85% NaCl, and 100% methanol). Embryos were stored in 100% methanol at -20 °C. Next, we rehydrated embryos through a graded series of 75%, 50%, 25% methanol/0.85% NaCl solutions for 5 min at room temperature. Embryos were next washed twice in 1x PBS with 0.1% Tween-20 (PBST) and treated with 10 mg/mL proteinase K in 1x PBST for 10 min (E8.0, E9.5) or 30 min (E10.5, E11.5, and E12.5). Samples were fixed again in 4% paraformaldehyde/0.2% glutaraldehyde in PBST for 20 min at room temperature, and washed twice in 1x PBST. We then incubated samples in pre-hybridization solution (50% formamide, 5× saline–sodium citrate (SSC) pH 4.5, 50 µg/ml yeast RNA, 1% SDS, 50 µg/ml heparin, and nuclease-free ddH$_2$O) for 1 h at 68 °C, and then incubated samples in 500 ng/mL of *Firre* antisense riboprobe at 68 °C for 16 h. Post hybridization, samples were washed in stringency buffers and incubated in 100 µg/mL RNaseA at 37 °C for 1 h. Next, samples were washed in 1× maleic acid buffer with 0.1% Tween-20 (MBST) and incubated in Roche Blocking Reagent (Roche, 1096176) with 10% heat-inactivated sheep serum (Sigma, S2263) for 4 h at room temperature. We used an anti-digoxigenin antibody (Roche, 11093274910) at 1:5000 and incubated the samples for 18 h at 4 °C. Samples were washed eight times with MBST for 15 min, five times in MBST for 1 h, and then once in MBST for 16 h at 4 °C. Samples were washed 3x for 5 min at room temperature with NTMT solution (100 mM NaCl, 100 mM Tris–HCl (pH 9.5), 50 mM MgCl$_2$, 0.1% Tween-20, 2 mM levamisole). The in situ hybridization signal was developed by adding BM Purple (Roche, 11442074001). After the colorimetric development, samples were fixed in 4% paraformaldehyde and cleared through a graded series of glycerol/1X PBS and stored in 80% glycerol. Embryos were imaged with a Leica M216FA stereo-microscope (Leica Microsystems) equipped with a DFC300 FX digital imaging camera in Leica Application Suite v2.3.4 R2.

**RNA-seq in embryonic tissues preparation and analysis**. For WT and Δ*Firre* RNA-seq in embryonic tissues, we dissected tissues (forebrain, midbrain, heart, lung, liver, forelimb, hindlimb, and pre-somitic mesoderm) from E11.5 embryos (44–48 somites) that were collected from matings between either male WT and female *Firre*$^{+/−}$ or male Δ*Firre* and female *Firre*$^{+/−}$ mice. Tissues were immediately homogenized in Trizol (Invitrogen), and the total RNA was isolated using RNeasy mini columns (Qiagen) on a QIAcube (Qiagen). Samples were genotyped for the WT, Δ*Firre*, and sex alleles. For each tissue, we generated the following libraries: WT male (n = 3), WT female (n = 3), Δ*Firre* male (n = 3), and Δ*Firre* female (n = 3). Poly(A)+ RNA-seq libraries were constructed using TruSeq RNA Sample Preparation Kit v2 (Illumina). The libraries were prepared using 500 ng of the total RNA as input, with the exception of the lung (200 ng) and the pre-somitic mesoderm (80 ng), and with a 10-cycle PCR enrichment to minimize PCR artifacts. The indexed libraries were pooled in groups of six, with each pool containing a mix of WT and Δ*Firre* samples. Pooled libraries were sequenced on an Illumina HiSeq 2500 in rapid-run mode with paired-end reads.

Reads were mapped to the mm10 mouse reference genome using TopHat v2.1.1 with the flags: "--no-coverage-search --GTF gencode.vM9.annotation.gtf", where this GTF is the Gencode vM9 reference gene annotation available at gencodegenes.org. Cufflinks v2.2.1 was used to quantify gene expression and assess the statistical significance of differences between conditions. Cuffdiff was used to independently compare the WT and Δ*Firre* samples and from each tissue and sex, and genes with FDR ≤0.05 were deemed significant (Supplementary Data 1–8). Gene Ontology analysis was performed using genes found significantly dysregulated in four or more embryonic tissues using FuncAssociate 3.0[58].

**RNA-seq in CLPs preparation and analysis**. We isolated CLPs (Lin⁻Sca1$^{lo}$cK-it$^{lo}$IL7Rα$^+$) by fluorescence-activated cell sorting (FACS) from mice 27–32 weeks old: WT (n = 4, mean age = 31 weeks), Δ*Firre* (n = 4, mean age = 30.4 weeks), and *Firre*$^{rescue}$ +dox (n = 4, mean age = 29.3 weeks). CLPs were directly sorted into TRIzol. RNA was isolated using RNeasy micro columns (Qiagen, 74004) on a QIAcube (Qiagen), and we quantified the concentration, and determined the RNA integrity using a BioAnalyzer (Agilent). Poly(A)+ RNA-seq libraries were constructed using CATS RNA-seq kit v2 (Diagenode, C05010041). Pooled libraries

were sequenced on an Illumina HiSeq 2500 in rapid-run mode with paired-end reads.

The adapter-trimmed reads were mapped to the mm10 mouse reference genome using TopHat v2.1.1 with the flags: "--no-coverage-search --GTF gencode.vM9.annotation.gtf" (Gencode vM9 reference gene annotation). FeatureCounts v1.6.2 and DESeq2 v1.14.1 were used to quantify gene expression and assess the statistical significance of differences between conditions[59],[60] and the P-value of comparisons were empirically calculated by using fdrtools v1.2.15 (ref.[61]). Genes with an FDR ≤0.1 were deemed significant in a comparison between wild-type and Δ*Firre* (Supplementary Data 9) and genes with an FDR ≤0.1 in the Δ*Firre* and *Firre*$^{rescue}$ comparison were deemed significant (Supplementary Data 10). Gene Ontology analysis was performed on significantly differentially regulated genes (FDR ≤0.1) in the WT and Δ*Firre* CLP comparison using PANTHER[62].

**qRT-PCR**. Embryonic tissues or cells were homogenized in Trizol (Invitrogen), and total RNA was isolated using RNeasy mini columns (Qiagen) on a QIAcube (Qiagen). In total, 300 ng of the isolated RNA was used as input to synthesize cDNA (SuperScript IV VILO Master Mix, Invitrogen, 11756050). Primers used qRT-PCR experiments: F_b-act: GCTGTATTCCCCTCCATCGTG, R_b-act: CACGGTTGGCCTTAGGGTTCAG; F_Firre: AAATCCGAGGACAGTCGAGC, R_Firre: CCGTGGCTGGTGACTTTTTG. Experiments were performed on a Viia7 (Applied Biosciences). qRT-PCR data were analyzed by the ΔΔCt method[63].

**Distribution of *Firre* expression across wild-type tissues**. For each of the eight WT embryonic tissues, FPKM estimates of all protein-coding or noncoding genes (biotypes selected: protein coding, lincRNA, and processed transcript) were aggregated and filtered for expression > 1 FPKM. Density and expression plots were generated using ggplot2_2.2.1 (geom_density()) in R version 3.3.3.

**Flow-cytometry analysis**. Zombie Aqua Fixable Viability Kit (Biolegend, 423101) was used as a live-dead stain, and used according to the manufacturer's protocol. For cell analysis of the peripheral blood, thymus, and bone marrow, age and sex-matched mice were used. Peripheral blood was collected by cardiac puncture or by tail vein collection into 10% by volume 4% citrate solution. The following antibodies were added (1:100) to each sample and incubated for 30 min at room temperature Alexa Fluor 700 anti-mouse CD8a clone 53-6.7 (Biolegend, 100730), PE/Dazzle-594 anti-mouse CD4 clone GK1.5 (Biolegend, 100456), APC anti-mouse CD19 clone 6D5 (Biolegend, 115512), Alexa Fluor 488 anti-mouse NK-1.1 clone PK136 (Biolegend, 108718), PE anti-mouse CD3 clone 17A2 (Biolegend, 100205), and TruStain FcX (anti-mouse CD16/32) clone 93 (1:50) (Biolegend, 101319). For competitive chimera experiments, Pacific Blue anti-mouse CD45.2 clone 104 (Biolegend, 109820) and PerCP anti-mouse CD45.1 clone A20 (Biolegend, 110726) were also used. Red blood cells were then lysed for 15 min at room temperature using BD FACS Lysing Solution (BD, 349202). Cells were washed twice in 1x PBS with 1% BSA and then resuspended in 1% paraformaldehyde or 1x PBS with 0.2% BSA.

Thymi were collected and homogenized in ice cold PBS over a 40-micron filter. The cells were incubated with the following antibodies (1:100) for 30 min at room temperature: Alexa Fluor 488 anti-mouse CD25 clone PC61 (Biolegend, 102017), PE/Cy7 anti-mouse/human CD44 clone IM7 (Biolegend, 103030), PE anti-mouse TCR β chain clone H57-597 (Biolegend, 109208), APC anti-mouse/human CD45R/B220 clone RA3-6B2 (Biolegend, 103212), eFluor 450 anti-mouse CD69 clone H1.2F3 (Invitrogen/eBioscience, 48069182), Alexa Fluor 700 anti-mouse CD8a clone 53-6.7 (Biolegend, 100730), and PE/Dazzle-594 anti-mouse CD4 clone GK1.5 (Biolegend, 100456). Cells were washed twice in 1x PBS with 1% BSA and then resuspended in 1% paraformaldehyde.

Bone marrow was collected from both femurs and tibias (four bones total per mouse) by removing the end caps and flushing with DMEM (Gibco, 11995-073) containing 5% fetal bovine serum (FBS) (Gibco, 26140079) and 10 mM EDTA. Cells were then pelleted, resuspended, and passed through a 70-micron filter. The resulting single-cell suspension was then incubated with the following antibodies (1:100) for 60 min on ice: Alexa Fluor 700 anti-mouse CD16/CD32 clone 93 (Invitrogen/eBioscience, 56-0161-82), PE/Cy7 anti-mouse CD127 (IL-7Rα) clone A7R34 (Biolegend, 135014), Alexa Fluor 488 anti-mouse CD117 (c-Kit) clone 2B8 (Biolegend, 105816), PE/Dazzle-594 anti-mouse Ly-6A/E (Sca1) clone D7 (Biolegend, 108138), APC anti-mouse CD34 clone HM34 (Biolegend, 128612), PE anti-mouse CD135 clone A2F10 (Biolegend, 135306), and Pacific Blue anti-mouse Lineage Cocktail (20 µL per 1×106 cells) clones 17A2/RB6-8C5/RA3-6B2/Ter-119/M1/70 (Biolegend, 133310). Red blood cells were lysed for 15 min at room temperature using BD FACS Lysing Solution (BD, 349202) or BD Pharm Lyse (BD, 555899). Cells were washed twice in 1x PBS with 1% BSA and then resuspended in 1% paraformaldehyde or 1x PBS with 0.2% BSA.

Flow cytometry was performed on a LSR-II (BD). To enumerate cell populations, 50 µL of CountBright Absolute Counting Beads (Invitrogen, C36950) was added to relevant bone marrow and thymus samples. Gating of cell populations was performed using FlowJo 10.4.1 software (Treestar) using the following criteria (applied to live singlets) (Supplementary Fig. 10): CD4 T cells (CD3+CD4+CD8-CD19-); CD8 T cells (CD3+CD8+CD4-CD19-); NK cells (NK1.1+CD19-CD3-); B cells (CD19+CD3-); double negative (DN) (B220-CD4-CD8-CD25$^{var}$CD44$^{var}$); double positive (DP) (CD4+CD8+B220-); single positive

(SP) (CD8, CD8+CD4-B220-); single positive (SP) (CD4, CD4+CD8-B220-); hematopoietic stem cells (HSC) (LSK [Lin-], Sca1+, c-Kit+)-CD34+-CD135-); multipotent progenitors (MPP) (LSK[CD34+CD135+]); common lymphoid progenitors (CLP) (Lin-Sca1$^{lo}$c-Kit$^{lo}$IL7Rα$^+$); and common myeloid progenitors (CMP) (LK[CD34+CD16/32-]). Negative gates were set using fluorescence-minus-one controls (FMO).

**Competitive HSC transplant assay.** Bone marrow from age- and sex-matched mice was collected and pooled with like genotypes (as described in flow-cytometry analysis section) from mice that were 8–9 weeks in age: PepBoy/CD45.1 ($n = 3$ females per experiment; mean age = 9 weeks) (Jackson Laboratory, 002014), *Firre* WT/CD45.2 ($n = 3$ females per experiment mean age = 8.9 weeks), and Δ*Firre*/ CD45.2 ($n = 3$ females per experiment; mean age = 8.6 weeks). Bone marrow was lineage-depleted according to the manufacturer's protocol (MiltinyiBiotec, 130-042-401), and cell marker surface staining was performed (as described for bone marrow in the Flow-cytometry analysis section). Red blood cells were then lysed for 15 min at room temperature using BD Pharm Lyse (BD, 555899). Cells were washed twice in 1x Hank's balanced salt solution (HBSS) (Gibco, 14025092) with 5% FBS and 2 mM EDTA. We then double-sorted lineage-depleted cells for an HSC-enriched population (Live, Lin-Sca1$^+$c-Kit$^+$CD34$^+$-CD135$^-$) into HBSS with 2% FBS using FACS (BD Aria). Recipient mice, PepBoy/CD45.1 ($n = 10$ males per experiment; mean age = 8.6 weeks), were lethally irradiated using a split 9.5γ split dose (3 h apart). *Firre* WT and Δ*Firre* HSCs were separately mixed at a 1:1 ratio with PepBoy/CD45.1 HSCs. In total, 100 μl containing 4000 mixed cells were transplanted by retro-orbital injection using 30-gauge insulin syringes (BD, 328411) into lethally irradiated recipients. Forty-eight hours post transplant, 100,000 helper marrow cells from male PepBoy/CD45.1 were transplanted by retro-orbital injection into each experimental PepBoy/CD45.1 male recipient mouse. After transplantation, mice were maintained on antibiotic-containing water (0.4% sulfamethoxazole/0.008% trimethoprim) for 4 weeks, and then switched to automatic watering of reverse osmosis deionized water chlorinated at 2 ppm.

**MEF preparations and culture.** We generated *Firre* WT, *Firre* knockout, and *Firre*$^{rescue}$ MEFs at E13.5 from intercrosses between male *Firre*$^{-/y}$ with female *Firre*$^{+/-}$ and male *Firre*$^{rescue}$ with female *Firre*$^{-/-}$. Individual embryos were dissected into 1x phosphate-buffered saline (PBS) and were eviscerated, and the head, forelimbs, and hindlimbs were removed. Embryo carcasses were placed into individual 6 cm$^2$ tissue culture plates containing 1 mL of pre-warmed 37 °C TrypLE (Thermo Fisher, 12604013) and were incubated at 37 °C for 20 min. Embryos were dissociated by gently pipetting using a P1000 tip and MEF media was added. Cells were cultured for 5–7 days, and cryostocks of individual lines were generated. Subsequent experiments were performed from thaws from the cryostocks up to passage 3. MEFs were genotyped for *Firre* WT, knockout, rtTA, tg(*Firre*), and sry alleles. MEF culture media: 1x Dulbecco's modified Eagle's medium (DMEM) (Invitrogen 11965-118), 10% FBS (Gibco, 10082139), L-glutamine (Thermo Fisher, 25030081), and penicillin/strepto-mycin (Thermo Fisher, 15140122).

**Firre FISH probe design and RNA FISH.** *Firre* oligo probes were designed using Primer3 (http://frodo.wi.mit.edu/primer3/), and synthesized by Integrated DNA Technologies. After Amine-ddUTP (Kerafast) was added to 2 pmol of pooled oligos by terminal transferase (New England Biolabs), oligos were labeled with Alexa647 NHS-ester (Life Technologies) in 0.1 M sodium borate. *Firre* RNA FISH was performed on *Firre* WT, Δ*Firre*, and *Firre*$^{rescue}$ MEFs which were plated at a density of 50,000 cells per well onto round glass coverslips in a 24-well plate. *Firre*$^{rescue}$ MEFs were cultured with either 2 μg/mL dox (Sigma, D9891) or vehicle (ddH$_2$O) for 24 h. Replicate wells were processed for either RNA FISH or to isolate RNA for *Firre* induction analysis by qRT-PCR. For RNA FISH, cells grown on glass coverslips were rinsed with 1x PBS and then fixed in 4% paraformaldehyde for 10 min at room temperature. Cells were permeabilized with 0.5% Triton X-100 in 1x PBS for 3 min at room temperature and then washed two times with 1x PBS for 5 min at room temperature. Cells were then dehydrated in a series of increasing ethanol concentrations. Six labeled oligo probes were added to hybridization buffer containing 25% formamide, 2X SSC, 10% dextran sulfate, and 1 mg/mL yeast tRNA. RNA FISH was performed in a humidified chamber at 42 °C for 4 h. Cells were washed three times in 2× SSC, and then were mounted for wide-field fluorescent imaging or dehydrated for STORM imaging. Nuclei were counter-stained with Hoechst 33342 (Life Technologies). The following pooled oligos against *Firre* were used: (1) AGCAGCAAATCCCAGGGGCC, (2) TTCCTCATTCCCCTTCTC CTGG, (3) CCCATCTGGGTCCAGCAGCA, (4) ATCAGCTGTGAGTGCCTTGC, (5) TCCAGTGCTTGCTCCTGATG, (6) GCCATGGTCAAGTCCTGCAT.

**Firre DNA/RNA and Xist RNA co-FISH.** Primary MEF cells were trypsinzed and cytospun to glass slides. After brief air drying, cells were incubated in PBS for 1 min, CSK/0.5% Triton X-100 for 2 min on ice, and CSK for 2 min on ice. Cells were fixed in 4% formaldehyde in PBS for 10 min at RT and washed twice in PBS. After dehydrated through series of EtOH washes, cells were subject to hybridization at 37 °C O/N with denatured digoxigenin-labeled *Xist* probe (50% formamide, 2× SSC, 10% dextran sulfate, 0.1 mg/mL CoT1 DNA). Cells were washed in 50% formamide, 2× SSC at 37 °C, and in 2× SSC at RT, three times each. RNA FISH

signal was detected by incubating FITC-labeled anti-digoxigenin antibody (Roche) in 4× SSC, 0.1% Tween-20 at 37 °C for 1 h and followed by washing in 4× SSC, 0.1% Tween-20 at 37 °C three times. Cells were fixed again in 4% formaldehyde in PBS for 10 min and washed twice in PBS. Cellular RNAs were removed by RNase A (Life Technologies) in PBS at 37 °C. After dehydrated through series of EtOH, cells were sealed in hybridization buffer (50% formamide, 2× SSC, 10% dextran sulfate, 0.1 mg/mL CoT-1 DNA) containing Cy3-labeled *Firre* probe (Fosmid WI-755K22). Chromosomal DNA and probes were denatured at 80 °C for 15 min and allowed to renature by cooling down to 37 °C O/N. Cells were washed in 50% formamide, 2× SSC at 37 °C and in 2× SSC at RT, three times each. Nuclei were counter-stained with Hoechst 33342 (Life Technologies). Imaging was performed on a Nikon 90i microscope equipped with a 60X/1.4 N.A. VC objective lens, Orca ER camera (Hamamatsu), and Volocity software (Perkin Elmer). All probes were prepared by nick translation using DNA polymerase I (New England Biolab), DNase I (Pro-mega), and Digoxigenin-dUTP (Roche), or Cy3-dUTP (Enzo Life Sciences).

**Skeletal preparations.** WT and Δ*Firre* E18.5 embryos were dissected and eviscerated. Samples were fixed in 100% ethanol for 24 h at room temperature. Embryos were then placed in 100% acetone for 24 h at room temperature and then incubated in staining solution (0.3% alcian blue 8GS (Sigma) and 0.1% Alizarin Red S (Sigma) in 70% ethanol containing 5% acetic acid) for 3 days at 37 °C. Samples were rinsed with distilled water and then placed in 1% potassium hydroxide at room temperature for 24 h. Samples were cleared in a series of incubations with 1% potassium hydroxide in 20%, 50%, and 80% glycerol. Skeletal preparations were placed in 80% glycerol/1x PBS and imaged using a Nikon D7000 camera with a Nikon 28-105 macro lens. Images were captured in the Nikon Electronic Format (NEF), and were processed in the Adobe Camera Raw plugin where exposure and temperature adjustments were applied to all like images.

**Reporting summary.** Further information on research design is available in the Nature Research Reporting Summary linked to this article.

## Data availability

RNA-sequencing data is available at Gene Expression Omnibus under the accession numbers GSE125683 and GSE83631. The data that support the findings of the study are available from the corresponding author upon request.

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

## Acknowledgements

We thank Dr. Martin Sauvageau for providing a *Firre* clone to generate a riboprobe; Dr. Diana Sanchez for assistance in the mouse facility; Dr. Susan Carpenter, Dr. Kate Pritchett-Corning, and Elektra Robinson for discussions on the LPS study; Joyce LaVecchio and Silvia Ionescu in the HSCRB flow cytometry core for FACS assistance; the Harvard Bauer Core for sequencing; Dr. Marta Melé for initial optimization for RNA-seq analysis; Dr. Rasim Barutcu for discussion on the manuscript; and Dr. Laurie Chen and Dr. Lin Wu at the Harvard Genome Modification Facility. This research was supported by the National Institutes of Health (NIH) General Medical Sciences postdoctoral fellowship award 1F32GM122335-01A1 (to J.P.L.) and support from NIH National Heart, Lung, and Blood Institute T32HL007893; NIH postdoctoral fellowship F32AG050395 (to J.M.G.); The Wellcome Trust Intermediate Clinical Fellowship (105920/Z/14/Z) (to J.C.L); Howard Hughes Medical Institute (to R.A.F); NIH RO1 AG048917 and the Dean's Initiative Award Program for Innovation Grants in the Basic and Social Sciences (to A.J.W.); and the Institute of Mental Health grant R01MH102416-03 and the NIH Institute of General Medical Sciences grant P01GM099117 (to J.L.R.) J.L.R is the Leslie Orgel Professor of RNA Science and HHMI Faculty Scholar.

## Author contributions

Study conceptualization and design: J.P.L., J.C.L., and J.L.R.; *Firre* ES cell targeting: A.W., J.H., and R.A.F.; transgenic mice generation and mouse husbandry, N.C. and J.P.L.; Immunophenotyping experiments: J.P.L., J.C.L., and J.M.G.; competitive chimera design and analysis: J.M.G., J.P.L., A.J.W.; endotoxic shock experiments: J.P.L., N.C., and C.G.; RNA-sequencing design and analysis: T.H., J.P.L., C.G., W.M., and A.G.; RNA FISH for *Firre*: H.S. and J.T.L.; funding and supervision: A.J.W. and J.L.R; writing the paper: J.P.L., J.C.L., and J.L.R. with input from all of the authors.

## Competing interests

The authors declare no competing interests.
