## [Peer Review File · Nature Communications]

Reviewers' comments:

Reviewer #1 (Remarks to the Author):

The manuscript by Rinn and colleagues reports a deep functional exploration of the FIRRE lncRNA. Despite early reports identifying this lncRNA, and cataloging several intriguing properties about this locus and RNA, there were many unknown questions about what, if anything, this lncRNA actually does. This study provides a detailed and thorough answer to this question by demonstrating clear physiological effects in an animal model. Moreover, the demonstration that they can rescue these observed effects by expressing the FIRRE lncRNA in trans both adds confidence to the veracity of the observed phenotype and also demonstrates an essential role for the FIRRE lncRNA (rather than DNA locus) in mediating these effects. This later statement is a key advance in its own right since there has been much uncertainty about whether FIRRE might simply represent an important DNA element (given its unusual structural properties and CTCF binding properties). This paper definitively answers this question and adds FIRRE to the still short list of bona fide functional lncRNAs that both have clear functional roles in vivo and that clearly act as a functional RNA molecule. I have no doubt that this paper will be of great interest to the lncRNA field broadly and, in my opinion, will set an important standard for the field for how to rigorously define lncRNA function.

I believe this paper should be published in Nature Communications.

Reviewer #2 (Remarks to the Author):

In this manuscript by Rinn and colleagues, new genetic models of lncRNA Firre loss- and gain-of-function are described and used to uncover a potential role for this lncRNA in T cell development. The main strength of the paper is the description of new genetically engineered mouse models which will be useful for investigating the functions of this lncRNA in vivo. In addition, the quality of the data that is presented in the manuscript is high. However, there are important weaknesses in the work which should, in my opinion, preclude its publication in a journal targeted to a broad readership such as Nature Communications.

First, the lethality of LPS in Firre overexpressing mice is of unclear physiologic significance (Figure 2e-f). Since there is no corresponding reciprocal phenotype (i.e. resistance to a lethal dose of LPS) in the Firre KO, this phenotype could be a result of ectopically expressing the RNA in a non-physiologic cell type (i.e. where it is not normally expressed) or due to expressing it at supraphysiologic levels (the relevant cell type is not identified nor is the level of Firre expression determined in it). In addition, one would want to see a phenotype like this reproduced in at least two independent transgenic lines since the transgene is randomly integrated into the genome at an unknown location. Demonstrating the absence of LPS-induced lethality in Firre-OE mice not treated with dox would help mitigate this specific concern, but it is standard practice to examine independent transgenic lines. Overall, these results are preliminary at best and are of unclear significance for understanding the function of the endogenous lncRNA.

The second major weakness is simply the underwhelming nature of the hematopoietic phenotype. In some experiments, the reduction in CLPs is ~2-fold (Fig 3b), whereas in other experiments, the reduction looks more like 20% (Fig. 4b). Effects on CD4/CD8 cells similarly vary across a similar range. No analysis of whether these changes in T cell numbers has any functional consequences on the immune system is provided. Without any mechanistic understanding of how the lncRNA mediates these effects, I question whether the demonstration of a phenotype as subtle as this is appropriate for publication in Nature Communications.

Other minor points:

1. Figure 1h (RNA FISH of MEFs) should be complemented with qRT-PCR measurements of Firre expression in MEFs of these genotypes.

2. Line 92-94: "Interestingly, GO analysis of the commonly dysregulated genes showed that deletion of the Firre locus affected genes involved in blood cell function and development (Fig. 2C)." This is not a very accurate description of the GO analysis, which is more consistent with effects on hemoglobin regulation specifically (not blood cell function) and general developmental processes rather than hematopoietic development.

3. Lines 121-122: "There was no block in thymic development, as normal proportions of cells were observed at each developmental stage (Extended Data Fig. 4)..." This is confusing as written because there appears to be a modest reduction in DN and DP cells in Firre KO thymus in this figure.

4. Lines 188-189: "In this study, we identified a central RNA-based role for the Firre locus during hematopoiesis." Although the authors may not mean this, this reads as suggesting that Firre has a central role in hematopoiesis, which would certainly be an overstatement based on the results presented here. Suggest re-phrasing.

Reviewer #3 (Remarks to the Author):

In this study, the authors generated 3 in vivo animal models to explain the function of lncRNA transcribed from X-chromosome Firre locus. lncRNAs are recognized as an important class of molecules in cellular function and analysis of their in vivo phenotypes is currently of high interest and importance. Using genetically modified animal models for loss-of-function, gain-of-function, and rescue-function for Firre, the authors showed lymphocytic lineage-specific defects during hematopoiesis and an impaired survival defect upon LPS challenge. These data are interesting and the experiments seem appropriate, however, more repeat experiments, inclusion of more controls in certain experiments, as well as more quantification are necessary in order to strengthen the conclusions the authors draw in this paper. These comments are further described below.

Major points:

1. What is the number of replicates for Fig 2f? The authors may want to add repeat experiments and increase the number of FirreOE-LPS mice and include FirreOE-saline as a control.

2. What is the number of replicates for Fig3b, c, extended Fig4 and extended Fig5?

3. Fig3b, c and Fig 4b seem to be taken from different contexts. Fig3b, c are whole bone marrow and Fig 4b is after lineage depletion and not whole bone marrow. Additionally, Fig4c is blood not bone marrow. This makes comparing the results across the figures challenging. Therefore, the authors may want to repeat the bone marrow analysis without lineage depletion for Fig4b and add blood analysis for Fig 3c.

4. The claim that the Firre is a trans-acting RNA molecule is not supported by the data presented. In order to test this claim, the authors could perform RNA-co-FISH in the Firre KO and rescue settings.

5. Is Firre expressed in the nucleus and/or cytoplasm? In Fig 1h, after rescue Firre expression seems to change from the nucleus to the cytoplasm. The authors may want to add quantification and discuss the differences between different cellular localizations and functions in the discussion section.

Minor:

1. The authors used both mm9 (line 215) and mm10 (line 282) in the manuscripts. It would be clearer to use only one mouse genome.

2. Fig 1d. The Firre locus (red arrow head on top) and deleted region (grey line under the

ideogram) do not line-up very well. Adding genome coordinates for the deleted region with a genome scale bar would be helpful.

3. Authors need to correct the year and title of the reference paper (line 99, 465). (2018) The full title is "The NF- κ B-Responsive Long Noncoding RNA Firre Regulates Posttranscriptional Regulation of inflammatory Gene Expression through interacting with hnRNPU."

4. There is a typo in line 220 ("matted").

5. It would be helpful if authors can add discussion of Firre expression levels in hematopoietic system disorders including malignancies.

Below we present a point-by-point response to the reviewers regarding the manuscript, “The *Firre* locus produces a trans-acting RNA molecule that functions in hematopoiesis.” The reviewer comments were helpful in adding additional lines of evidence to help bolster our results and conclusions. Their insightful suggestions also helped to better contextualize our results. We believe that our manuscript has improved significantly with the comments and suggestions from the reviewers. Our replies to the reviewer comments are below in blue text.

Reviewers' comments:

Reviewer #1 (Remarks to the Author):

1. The manuscript by Rinn and colleagues reports a deep functional exploration of the FIRRE lncRNA. Despite early reports identifying this lncRNA, and cataloging several intriguing properties about this locus and RNA, there were many unknown questions about what, if anything, this lncRNA actually does. This study provides a detailed and thorough answer to this question by demonstrating clear physiological effects in an animal model. Moreover, the demonstration that they can rescue these observed effects by expressing the FIRRE lncRNA in trans both adds confidence to the veracity of the observed phenotype and also demonstrates an essential role for the FIRRE lncRNA (rather than DNA locus) in mediating these effects. This later statement is a key advance in its own right since there has been much uncertainty about whether FIRRE might simply represent an important DNA element (given its unusual structural properties and CTCF binding properties). This paper definitively answers this question and adds FIRRE to the still short list of bona fide functional lncRNAs that both have clear functional roles in vivo and that clearly act as a functional RNA molecule. I have no doubt that this paper will be of great interest to the lncRNA field broadly and, in my opinion, will set an important standard for the field for how to rigorously define lncRNA function.

I believe this paper should be published in Nature Communications.

We are grateful for these comments and the reviewer’s opinion that our manuscript is “detailed and thorough” and should be of “great interest”.

Reviewer #2 (Remarks to the Author):

In this manuscript by Rinn and colleagues, new genetic models of lncRNA *Firre* loss- and gain-of-function are described and used to uncover a potential role for this lncRNA in T cell development. The main strength of the paper is the description of new genetically engineered mouse models which will be useful for investigating the functions of this lncRNA in vivo. In addition, the quality of the data that is presented in the manuscript is high. However, there are important weaknesses in the work which should, in my opinion, preclude its publication in a journal targeted to a broad readership such as Nature Communications.

1. First, the lethality of LPS in *Firre* overexpressing mice is of unclear physiologic significance (Figure 2e-f). Since there is no corresponding reciprocal phenotype (i.e. resistance to a lethal dose of LPS) in the *Firre* KO, this phenotype could be a result of ectopically expressing the RNA in a non-physiologic cell type (i.e. where it is not normally expressed) or due to expressing it at

supraphysiologic levels (the relevant cell type is not identified nor is the level of Firre expression determined in it).

---“the lethality of LPS in Firre overexpressing mice is of unclear physiologic significance (Figure 2e-f)”---

We agree with the reviewer that providing information on what is known about Firre overexpression from the literature would be helpful to contextualize our findings.

To clarify, this study is the first to assess the *in vivo* physiological significance of Firre loss- and gain-of function in the context of an endotoxin challenge (LPS). The rationale of why we used an endotoxin challenge in Firre loss- and gain-of-function models is because two recent studies (using cell-based assays) showed that Firre is a NF- κ B regulated and LPS-responsive lncRNA that can modulate the response key genes involved in innate immunity (Lu et al., 2017 PMID: 28993514 and Zhang et al., 2018 PMID: 29715458). As the main objective of this study, we reasoned that an endotoxin challenge could be used as a model to determine potential DNA- and RNA-mediated effects of Firre *in vivo*.

In this study, we found that serum cytokines that have key roles in innate immune function (such as TNF- α) were significantly increased in Firre-OE (induced) mice upon exposure to LPS (**Fig. 2F and we performed new experiments in Fig. 2G**). We then hypothesized that the increase in serum cytokines in Firre-OE (induced) mice could have an impact on survival (**we performed new experiments for the LPS survival study and discuss in point #2**).

In addition, Reviewer #3 also suggested to provide discussion on what is known about Firre in the context of disease and overexpression. To address both reviewer’s concerns, we reviewed the literature and comment on Firre overexpression in human disease and immune regulation in the **results and discussion sections** in order to better contextualize our findings:

(lines 196-203): “Cell-based approaches have shown that Firre can be transcriptionally upregulated upon exposure to lipopolysaccharide (LPS) and can modulate the levels of inflammatory genes in human colorectal adenocarcinoma cells (SW480)³⁴, mouse macrophage cells (RAW264.7)³⁴, as well as in an injury model using cultured primary rat microglial cells³⁷. Thus, we hypothesized that dysregulation of Firre might alter the inflammatory response *in vivo* and reasoned that investigating the inflammatory response in Firre loss- and gain-of function mice could provide insight into the DNA- and RNA-mediated effects of Firre.”

(lines 381-387): “Second, we observed that overexpression of *Firre* RNA in an endotoxic shock model potentiated the innate immune response *in vivo*. Consistent with our findings, independent studies have found that overexpression of *Firre* RNA in different cellular contexts can modulate genes in the innate immune response. For example, the levels of select inflammatory genes, including IL12p40, are increased upon overexpression of *Firre* RNA followed by LPS stimulation in human SW480 cells³⁴. In addition, upon overexpression of *Firre* RNA in an injury model using cultured primary microglial cells, TNF- α levels are increased³⁷.”

(lines 388-392): However, we speculate that *Firre* could be important in setting functional thresholds for cells. For example, FIRRE is not only significantly increased in certain cancers including kidney renal cell carcinoma, lung squamous cell carcinoma, and colon adenocarcinoma³⁸, but high levels of FIRRE expression have been significantly

associated with more aggressive disease and poor survival in patients with large B-cell lymphoma³⁶.

---"Since there is no corresponding reciprocal phenotype (i.e. resistance to a lethal dose of LPS) in the Firre KO, this phenotype could be a result of ectopically expressing the RNA in a non-physiologic cell type (i.e. where it is not normally expressed) or due to expressing it at supraphysiologic levels"---

We agree with the reviewer that a limitation of the overexpression approach is that we do not know if the phenotype is due to increased levels in a cell type where *Firre* is normally expressed or de novo expression in a cell type where *Firre* is not normally expressed. We now acknowledge this important point in the **discussion section**:

(lines 388-392): "In the overexpression approach in this study, we note that the phenotype could be due to increased levels of *Firre* in a cell type where it is normally expressed or de novo expression in a cellular context where *Firre* is not normally found; perhaps mimicking disease contexts where *FIRRE* is found overexpressed."

2. In addition, one would want to see a phenotype like this reproduced in at least two independent transgenic lines since the transgene is randomly integrated into the genome at an unknown location. Demonstrating the absence of LPS-induced lethality in *Firre*-OE mice not treated with dox would help mitigate this specific concern, but it is standard practice to examine independent transgenic lines. Overall, these results are preliminary at best and are of unclear significance for understanding the function of the endogenous lncRNA.

--- "Demonstrating the absence of LPS-induced lethality in *Firre*-OE mice not treated with dox would help mitigate this specific concern"---

We agree with the reviewer that it is important to ensure that any observed phenotype in a transgenic animal is not due to disruption of the integration site. We thank the reviewer for the suggestion to use an 'uninduced' *Firre*-OE control group in order to control for the genetic background of the *Firre*-OE line. In the *Firre*-OE model, the transgene in the 'uninduced' control is not expressed without the addition of doxycycline within the food. If the observed phenotype were due to the disruption of the endogenous integration site, then the same phenotype should be observed irrespective of whether the transgene is expressed. Below, we discuss the results from three new complementary experiments that we think address the reviewer's specific concern and present the new results in **Fig. 2H, Fig. 2G, and Extd. Data Fig. 4** (accompanying text in **results section lines 195-228**).

First: We designed a new 6-day LPS survival study that included a ‘uninduced’ control group, increased the number of mice per genotype, and increased experimental replicates to two. We performed the LPS survival study using a preparation of LPS that specifically stimulates toll-like receptor 4 (TLR4) at a sub-lethal dose in WT on two independent cohorts of mice, consisting in total of: WT (n=30), Firre KO (n=18), Firre-OE (uninduced) (n=13), and Firre-OE (induced) (n=17), as well as a saline control group consisting of all genotypes (n=10).

Figure 2H. (NEW DATA) 6-day survival plot of mice injected with 5 mg/kg LPS

Consistent with the previous LPS survival study presented in the first submission of the manuscript, Firre-OE (induced) showed significant impaired survival compared to WT ($P < 0.0001$, Mantel-Cox) upon exposure to LPS. Further, as the reviewer suggested, the comparison between Firre-OE (induced) and Firre-OE (uninduced) mice allows for a direct comparison between the specific genetic background of Firre-OE mice. In this comparison, Firre-OE (induced) mice were also significantly more susceptible to LPS compared

to Firre-OE (uninduced) ($P = 0.0063$, Mantel-Cox) (Fig. 2H). Indeed, we do not observe the same phenotype between Firre-OE (uninduced) and Firre-OE (induced) mice. We interpret these data as that the increased lethality effect can be attributed to the overexpression of Firre RNA rather than the insertion site upon exposure to LPS.

Second: Because we used a new preparation of LPS that is specific to TLR4, we performed a new serum cytokine analysis in WT, Firre-OE (uninduced), and Firre-OE (induced) mice upon exposure to the TLR4-specific LPS preparation (Fig. 2G and text lines 203-214). In this new experiment, Firre-OE (induced) mice also showed significantly higher level of pro-inflammatory cytokines (Fig. 2G). This data is consistent with the original serum cytokine data presented in Fig. 2F that uses a broader-acting preparation of LPS.

Figure 2G. (NEW DATA) Serum cytokine analysis

Last: we designed an experiment to test whether simply overexpressing the Firre transgene from its random integration site causes increased pro-inflammatory serum cytokines. In these new results (Extended Data Figure 4 and text lines 210-214), we found that Firre-OE (induced) mice injected with saline do not have increased levels of pro-inflammatory cytokines relative to the control group also injected with saline (Extended Data Figure 4). These data demonstrate that

simply overexpressing transgenic Firre from its random integration site does not lead to an increase pro-inflammatory cytokine response.

Extended Data Figure 4. (NEW DATA) Serum cytokine analysis

---"Overall, these results are preliminary at best and are of unclear significance for understanding the function of the endogenous lncRNA."---

We believe these new experimental data (discussed above) – that include more independent cohorts, replicates, and controls – expand our findings beyond preliminary. The results from the new experiments provide additional evidence that in Firre-OE (induced) mice upon exposure to LPS, the lethality and increased pro-inflammatory cytokine response is mediated by expression of transgenic Firre RNA rather than by the integration of the Firre transgene.

3. The second major weakness is simply the underwhelming nature of the hematopoietic phenotype. In some experiments, the reduction in CLPs is ~2-fold (Fig 3b), whereas in other experiments, the reduction looks more like 20% (Fig. 4b).

--- "In some experiments, the reduction in CLPs is ~2-fold (Fig 3b), whereas in other experiments, the reduction looks more like 20% (Fig. 4b)."

We apologize to the reviewer for the confusion. In the first submission of the manuscript one figure panel showed the frequencies of CLPs from total bone marrow and the other one showed the frequencies of CLPs from lineage-depleted bone marrow. The different processing methods of (total vs lineage depleted) bone marrow accounts for the differences in frequencies of CLPs.

Reviewer #3 also mentioned the differences between the two previous figure panels, and suggested to include more replicate experiments using total bone marrow.

Figure 4B. (NEW DATA)
Frequencies of CLPs in total bone marrow from WT, KO, and Rescue

To address both of the reviewer's comments, we generated new experimental mouse cohorts and determined the frequency of CLPs in total bone marrow for WT (n=16), Firre KO (n=17), and Firre-rescue (induced) (n=15) mice. In these new data, we found that the frequency of CLPs is significantly decreased in Firre KO samples relative to WT (Fig. 4B). In addition, upon induction of Firre RNA from a transgene in the Firre KO background (Firre-rescue (induced)), the frequencies of CLPs are significantly higher in the Firre-rescue (induced) mice compared to Firre KO - essentially restored to the CLP frequency found in WT (Fig. 4B). This new data are now consistent with the experimental approach to assess CLP frequency presented in Fig. 3B. Further, this new data is also consistent with the rescue of CLP frequencies that we observed in lineage-depleted bone marrow samples -- we have moved the data to **Extended Data Fig. 8** and have updated the text in the **results section (lines 283-295)**.

4. Effects on CD4/CD8 cells similarly vary across a similar range. No analysis of whether these changes in T cell numbers has any functional consequences on the immune system is provided. Without any mechanistic understanding of how the lncRNA mediates these effects, I question whether the demonstration of a phenotype as subtle as this is appropriate for publication in Nature Communications.

We agree with the reviewer that investigating the functional consequences on CD4/CD8 T cells in Firre KO mice is an interesting question and warrants further investigation, but it is out of the scope of this study. We selected to use the CLP as a model to further investigate the physiological and molecular (DNA- and RNA- mediated) effects of Firre in a progenitor cell population. We reasoned that the CLP could be a useful model because this was the earliest hematopoietic defect identified by immunophenotyping and because Firre is highly expressed in this progenitor cell type compared to T cells.

To clarify, the main goal of our study was to determine DNA- and RNA-mediated effects of the Firre locus in vivo – if any. In this study we generated three new mouse models and identified a number of phenotypes in Firre loss- and gain-of-function mice – both physiological and molecular. We determined that modulating Firre directs cell-specific defects during hematopoiesis, potentiates the innate immune response upon exposure to LPS, and can restore a gene expression program a progenitor cell population – all of which have an RNA-based functional modality. Together, this study provides evidence that Firre can function as a trans-acting RNA in vivo, and we believe this key insight will be important to future work that investigate the potential role(s) of Firre in other biological contexts as well as in disease.

Other minor points:

1. Figure 1h (RNA FISH of MEFs) should be complemented with qRT-PCR measurements of Firre expression in MEFs of these genotypes.

We have quantified Firre RNA expression by qRT-PCR in MEFs from WT, Firre KO, Firre-rescue (uninduced) and Firre-rescue (induced) and have included this new data in **Fig. 1i** and **discussed in lines 170-171**. Briefly, we observe a ~2.7-fold increase in Firre RNA in the Firre-rescue MEFs (induced) compared to WT. As controls, we do not detect Firre expression in Firre KO or in uninduced Firre-rescue MEFs

Figure 1i. (NEW DATA) qRT-PCR for Firre

2. Line 92-94: “Interestingly, GO analysis of the commonly dysregulated genes showed that deletion of the Firre locus affected genes involved in blood cell function and development (Fig. 2C).” This is not a very accurate description of the GO analysis, which is more consistent with effects on hemoglobin regulation specifically (not blood cell function) and general developmental processes rather than hematopoietic development.

We agree with this point and thank the reviewer for making this suggestion. We have updated the text to more accurately reflect the GO terms in the **results section (lines 186-188)**: “gene ontology (GO) analysis of the commonly dysregulated genes showed that deletion of the Firre locus affected genes involved in hemoglobin regulation and general blood developmental processes (**Fig. 2C**).”

3. Lines 121-122: “There was no block in thymic development, as normal proportions of cells were observed at each developmental stage (Extended Data Fig. 4)...” This is confusing as written because there appears to be a modest reduction in DN and DP cells in Firre KO thymus in this figure.

We apologize for the confusion regarding this figure and the accompanying explanation in the text. In Extended Fig 5B the upper panels show the proportion of each cell type and the lower panels show the absolute cell number. The panels that the reviewer is referring to, which show a reduction in DN and DP cells, actually depict cell number - which is reduced in Firre KO mice. In fact, there is a general trend to a reduction in cell number for all of the populations shown - consistent with there being fewer cells entering the thymus, which in turn is consistent with a primary reduction in CLPs. In contrast, the proportion of each population (as a percentage of all live cells) is largely the same. These data are therefore consistent with the primary abnormality

being a reduction in CLPs, and suggest that thymic development is normal. We have re-worded the text in the **results section** to make this less confusing:

(lines 238-246) “There was no block in thymic development in Δ Firre mice, as normal frequencies of cells were observed at each developmental stage (Extended Data Fig. 5B, upper panels). However, we noticed that the absolute number of cells was generally lower in Δ Firre mice at every developmental stage, suggestive of a pre-thymic defect in progenitor development (Extended Data Fig. 5B, lower panels). Consistent with this, in the bone marrow compartment, we observed a significant reduction in both the frequency and number of common lymphoid progenitors (CLP) in Δ Firre mice ($P=0.0069$ and $P<0.0001$, respectively) (Fig. 3B and Extended Data Fig. 5C).”

We have also re-labelled the axes in **Extended Data Fig. 5** to more clearly indicate that the upper panels depicts cell frequency, “Frequencies of cells (% of live),” and the lower panels depict cell number, “Number of cells per μ L”

4. Lines 188-189: “In this study, we identified a central RNA-based role for the Firre locus during hematopoiesis.” Although the authors may not mean this, this reads as suggesting that Firre has a central role in hematopoiesis, which would certainly be an overstatement based on the results presented here. Suggest re-phrasing.

We thank the reviewer for this comment because we did not intend to say that Firre has a central role in hematopoiesis. We have corrected this in a few places throughout the manuscript:

(lines 107-108): our study provides evidence for a *trans*-acting RNA-based role for the *Firre* locus that has physiological importance for hematopoiesis and immune function.

(lines 365-366): “we identified that Firre produces a *trans*-acting lncRNA in a hematopoietic context”

Reviewer #3 (Remarks to the Author):

In this study, the authors generated 3 in vivo animal models to explain the function of lncRNA transcribed from X-chromosome Firre locus. lncRNAs are recognized as an important class of molecules in cellular function and analysis of their in vivo phenotypes is currently of high interest and importance. Using genetically modified animal models for loss-of-function, gain-of-function, and rescue-function for Firre, the authors showed lymphocytic lineage-specific defects during hematopoiesis and an impaired survival defect upon LPS challenge. These data are interesting and the experiments seem appropriate, however, more repeat experiments, inclusion of more controls in certain experiments, as well as more quantification are necessary in order to strengthen the conclusions the authors draw in this paper. These comments are further described below.

Major points:

1. What is the number of replicates for Fig 2f? The authors may want to add repeat experiments and increase the number of FirreOE-LPS mice and include FirreOE-saline as a control.

We thank the reviewer for this comment. Reviewer #1 also suggested additional controls and replicate experiments. Briefly, we have designed and performed a new LPS survival study which consisted of **two replicate experiments, increased the number of Firre-OE LPS mice from 5 to 17**, and has **added additional controls** including a Firre-OE LPS (uninduced) cohort to control for background effects as well as a Firre-OE (induced) mice included in the saline control group. Below, we discuss the results from three new complementary experiments that we think address the reviewers specific concern and present the new results in **Fig. 2H, Fig. 2G, and Extd. Data Fig. 4** (accompanying text in **results section lines 195-228**).

First: We designed a new 6-day LPS survival study that included a ‘uninduced’ control group, increased the number of mice per genotype, and increased experimental replicates to two. We performed the LPS survival study using a preparation of LPS that specifically stimulates toll-like receptor 4 (TLR4) at a sub-lethal dose in WT on two independent cohorts of mice, consisting in total of: WT (n=30), Firre KO (n=18), Firre-OE (uninduced) (n=13), and Firre-OE (induced) (n=17), as well as a saline control group consisting of all genotypes (n=10).

Figure 2H. (NEW DATA) 6-day survival plot of mice injected with 5 mg/kg LPS

Consistent with the previous LPS survival study shown in the first submission of the manuscript, Firre-OE (induced) showed significant impaired survival compared to WT ($P < 0.0001$, Mantel-Cox) upon exposure to LPS. Further, as the reviewer suggested, the comparison between Firre-OE (induced) and Firre-OE (uninduced) mice allows for a direct comparison between the specific genetic background of Firre-OE mice. In this comparison, Firre-OE (induced) mice were also significantly more susceptible to LPS compared to Firre-OE (uninduced) ($P = 0.0063$,

Mantel-Cox) (**Fig. 2H**). Indeed, we do not observe the same phenotype between Firre-OE (uninduced) and Firre-OE (induced), thus we interpret these data as that the increased lethality effect can be attributed to the overexpression of Firre RNA rather than the insertion site upon exposure to LPS.

Second: Because we used a new preparation of LPS that is specific to TLR4, we performed a new serum cytokine analysis in WT, Firre-OE (uninduced), and Firre-OE (induced) mice upon exposure to the TLR4-specific LPS preparation (**Fig. 2G and text lines 203-214**). In this new experiment, Firre-OE (induced) mice also showed significantly higher level of pro-inflammatory cytokines (**Fig. 2G**). This data is consistent with the original serum cytokine data presented in **Fig. 2F** that uses a broader-acting preparation of LPS.

Figure 2G. (NEW DATA) Serum cytokine analysis

Last: We designed an experiment to test whether simply overexpressing the Firre transgene from its random integration site causes increased pro-inflammatory serum cytokines. In these new results (**Extended Data Figure 4** and text lines 210-214), we found that Firre-OE (induced) mice injected with saline do not have increased levels of pro-inflammatory cytokines relative to the control group also injected with saline (**Extended Data Figure 4**). This data demonstrate that simply overexpressing transgenic Firre from its random integration site does not lead to an increase pro-inflammatory cytokine response.

Extended Data Figure 4. (NEW DATA) Serum cytokine analysis

2. What is the number of replicates for Fig3b, c, extended Fig4 and extended Fig5?

We have added replicate / number information in the figure legends and methods section.

3. Fig3b, c and Fig 4b seem to be taken from different contexts. Fig3b, c are whole bone marrow and Fig 4b is after lineage depletion and not whole bone marrow. Additionally, Fig4c is blood not bone marrow. This makes comparing the results across the figures challenging. Therefore, the authors may want to repeat the bone marrow analysis without lineage depletion for Fig4b and add blood analysis for Fig 3c.

We apologize to the reviewer for the confusion (reviewer #1 also mentioned the differences between the two previous figure panels). In the first submission of the manuscript one figure panel showed the frequencies of CLPs from total bone marrow and the other one showed the frequencies of CLPs from lineage-depleted bone marrow. The different processing methods of (total vs lineage depleted) bone marrow accounts for the differences in frequencies of CLPs.

To address both of the reviewer's comments, we generated new experimental mouse cohorts and determined the frequency of CLPs in total bone marrow for WT (n=16), Firre KO (n=17), and Firre-rescue (induced) (n=15) mice. In these new data, we

Figure 4B. (NEW DATA)
Frequencies of CLPs in total bone marrow from WT, KO, and Rescue

find that the frequency of CLPs are significantly decreased in Firre KO samples relative to WT (**Fig. 4B**). And upon induction of Firre RNA from a transgene in the Firre KO background (Firre-rescue (induced)), the frequencies of CLPs are significantly higher in the Firre-rescue (induced) mice compared to Firre KO - essentially restored to the CLP frequency found in WT (**Fig. 4B**). This new data are now consistent with the experimental approach to assess CLP frequency presented in **Fig. 3B**. Further, this new data is also consistent with the rescue of CLP frequencies that we observed in lineage-depleted bone marrow samples -- we have moved the data to **Extended Data Fig. 8** and have updated the text in the **results section (lines 283-295)**.

4. The claim that the Firre is a trans-acting RNA molecule is not supported by the data presented. In order to test this claim, the authors could perform RNA-co-FISH in the Firre KO and rescue settings.

We thank the reviewer for this comment and we agree that multiple approaches should be used to investigate the cis- and trans-based effects of lncRNA loci. We would like to further clarify some experimental results and highlight additional evidence to support the conclusion that Firre can function as a trans-acting RNA in vivo. We believe the following data in the manuscript supports the conclusion that Firre is a trans-acting RNA:

- a) We do not find evidence of a local (cis) role for Firre DNA or RNA in vivo. Deletion of Firre in 9 different biological contexts did not disrupt neighboring gene expression within 2 Mb window centered on the Firre locus (**Fig. 4G and Extended Data Fig. 9 and lines 324-342**). Thus any phenotypes observed in the Firre KO model are not likely do to a local (cis) role for Firre DNA or RNA.
- b) Specific to the reviewers comment, we have performed RNA FISH for Firre in Firre KO and Firre-Rescue (induced) MEFs (please see **Fig. 1H**, we provide further commentary on this result in point #4). We found that Firre is predominately distributed throughout the nucleus in WT MEFs, which is consistent with the notion that Firre has the potential to exert a regulatory role at sites away from the site where it is transcribed (trans-acting).
- c) We find evidence that the Firre lncRNA can exert an effect in trans. The decreased frequency of CLPs that we observed in Firre KO mice is restored to that of WT upon induction of Firre RNA from a transgene that is in the Firre KO background (Firre-rescue (induced)) (**Figure 4B and Extended Data Fig. 8, lines 269-298**).

- d) Further, RNA-seq analysis provided further evidence that Firre can function as a trans-acting RNA. Genes found significantly dysregulated in CLPs from Firre KO mice could be rescued upon induction of Firre RNA in the Firre-rescue model (**Fig. 4D, F and lines 300-325**).

Together, these four key pieces of data -- which have been used as criteria to define other trans-acting lincRNAs such as lincEPS (Atianand et al., 2016 PMID: 27315481 and reviewed in Kopp and Mendel 2018 PMID: 29373828) -- are consistent with a trans RNA based role for the Firre lincRNA in vivo.

5. Is Firre expressed in the nucleus and/or cytoplasm? In Fig 1h, after rescue Firre expression seems to change from the nucleus to the cytoplasm. The authors may want to add quantification and discuss the differences between different cellular localizations and functions in the discussion section.

We find that Firre RNA is predominately localized to the nucleus in WT MEFs (**Fig. 1H**). This localization pattern has also been observed in mouse embryonic stem cells and neuronal precursor cells (Hacisuleyman et al., 2014 PMID:24463464, Bergman et al., 2015 PMID: 26048247). However, FIRRE has also been reported to localize predominately in the cytoplasm in a human colon cell line (Lu et al., 2017 PMID: 28993514).

In this study, we performed RNA FISH for Firre in MEFs from Firre-rescue mice because they lack endogenous Firre, so we are only looking at RNA from the transgene. In this experiment we observe that Firre is distributed throughout both the nucleus and cytoplasm – we speculate this could be due to a threshold level mechanism for nuclear localized Firre, as HNRNPU has been previously shown to localize Firre RNA to the nucleus by independent laboratories (Hacisuleyman et al., 2014 PMID:24463464 and Lu et al., 2017 PMID: 28993514).

As also suggested by reviewer 1, we have performed quantification of Firre RNA (qRT-PCR) in WT, Firre KO, Firre-rescue (uninduced), and Firre-rescue (induced) MEF samples. This new data is incorporated into **Fig. 1I**. We have also commented on the different cellular localizations found for Firre RNA in the **results section**:

Figure 1I. (NEW DATA) qRTPCR for Firre

(lines164-173) “Firre RNA has been reported to be largely enriched in the nucleus of mouse embryonic stem cells (mESCs)^{23,47}, neuronal precursor cells³⁹, and HEK293 cells¹⁷, but also has been reported in the cytoplasm of a human colon cell line³⁴. Thus, we investigated the subcellular localization of Firre in the genetic models using RNA fluorescent in situ hybridization (RNA FISH). In contrast to Δ Firre MEFs, we detected

pronounced localization of Firre RNA in the nucleus of WT MEFs (**Fig. 1H**). In dox-treated Firre^{rescue} MEFs, which only produce Firre RNA from the transgene, we detected Firre RNA in both the nucleus and cytoplasm (**Fig. 1H**), which corresponded to approximately a 2.7-fold increase in Firre RNA relative to WT (**Fig. 1I**). Notably, the Firre^{rescue} transgenic model showed both nuclear and cytoplasmic localization of Firre, suggesting a threshold level control for nuclear localized Firre.”

Minor:

1. The authors used both mm9 (line 215) and mm10 (line 282) in the manuscripts. It would be clearer to use only one mouse genome.

We apologize for the confusion and have clarified the use of two different reference genomes. The data presented in **Fig. 1D** is in mm9 because the ChIP-seq data from ENCODE (via UCSC Genome Browser) was produced using mm9 and we generated a corresponding Firre WT and KO RNA-seq file for mm9 in order to match the ENCODE data. All other RNA-seq data and analyses presented in the paper were performed in mm10. We have added mm9 to the figure legend in **Fig. 1D** to be more clear and have updated the **methods section** accordingly.

2. Fig 1d. The Firre locus (red arrow head on top) and deleted region (grey line under the ideogram) do not line-up very well. Adding genome coordinates for the deleted region with a genome scale bar would be helpful.

We apologize for this error. We have re-aligned Firre and Xist with the ideogram and have added a scale bar to **Fig. 1D**, and have included the deletion coordinates in the **methods section**.

3. Authors need to correct the year and title of the reference paper (line 99, 465). (2018) The full title is “The NF-κB-Responsive Long Noncoding RNA Firre Regulates Posttranscriptional Regulation of inflammatory Gene Expression through interacting with hnRNP.”

We thank the reviewer for pointing this out and we have fixed the mistake in the citation.

4. There is a typo in line 220 (“matted”).

We thank the reviewer for pointing this out and we have fixed this mistake.

5. It would be helpful if authors can add discussion of Firre expression levels in hematopoietic system disorders including malignancies.

We thank the reviewer for this suggestion to help contextualize our results. We further searched the literature for information on Firre expression in malignancies as well as hematopoietic system disorders and malignancies and found that Firre is overexpressed in a number of cancers including, colorectal cancer (Li et al., 2017 PMID: 28731151 and Sun et al., 2017 PMID: 28427191), and in 3 of the 13 cancer types (kidney renal clear cell carcinoma, colon

adenocarcinoma, and lung squamous cell carcinoma) analyzed from The Cancer Genome Atlas (Yan et al., 2015 PMID: 26461095).

More specific to *Firre* in hematopoietic disorders, we found that in CUTLL1 cells (a model for T-cell acute lymphoblastic leukemia) *Firre* transcription is responsive to Notch pathway activation as well as Notch pathway inhibition (Durinck et al., 2014 PMID: 25344525). Further, we found that patients with diffuse large B-cell lymphoma (DLBCL) have a significant increase in *FIRRE* expression compared to healthy controls, and that high levels of *FIRRE* had a significant association with poor survival in DLBCL patients (Shi et al., 2019, PMID: 30739786).

We have added discussion of *Firre* expression in malignancies as well as hematopoietic system disorders to the **discussion section** in the manuscript:

(lines 388-393): “We speculate that *Firre* could be important in setting functional thresholds for cells. For example, *FIRRE* is not only significantly increased in certain cancers including, kidney renal cell carcinoma, lung squamous cell carcinoma, and colon adenocarcinoma³⁸, but high levels of *FIRRE* expression have been significantly associated with more aggressive disease and poor survival in patients with large B-cell lymphoma³⁶.”

REVIEWERS' COMMENTS:

Reviewer #2 (Remarks to the Author):

The authors have invested significant effort in addressing the comments of the reviewers and, overall, I am satisfied with the revisions and now support publication. I have a few final requests that I think could improve and/or clarify some of the points made in the manuscript.

1. At the end of the section that describes the LPS challenge experiments (line 230), the authors should add an additional sentence that explicitly states that endogenous Firre does not appear to be necessary for the normal inflammatory response to LPS challenge. This is clear from Figure 2f, which shows that WT and Firre KO mice produce equivalent cytokines after LPS administration.
2. Also regarding the LPS challenge experiment, it is usually recommended to include an rtTA only + Dox control (to rule out any unexpected effects of Dox treatment). If the authors have such data, it would strengthen their results if they added it to the manuscript. In the future, I recommend including this control.
3. In the experiments shown in Figure 4d-f (rescue of gene expression changes in CLPs) the authors emphasize that 78 of the 89 genes that were differentially expressed in Firre KO CLPs were reciprocally regulated in Firre-rescue CLPs. Nevertheless, the comparison of Firre-rescue to Firre KO revealed 4656 differentially expressed genes, raising concerns that they are not really seeing a specific rescue. 4656 genes probably represents 1/3 to 1/2 of all genes expressed in CLPs. Given this massive change in gene expression, it is less impressive that the small number of genes that are aberrantly expressed in Firre KO cells show some reversal of this effect. In the final manuscript, the authors should discuss the possibility that some of the apparent rescue of gene expression that they observed in Firre-rescue mice is due to the extremely widespread changes in gene expression observed in those animals rather than specific rescue of the 89 differentially expressed genes detected in Firre KO mice.
4. Table 1 is confusing. It is unclear what the expected number of mice of each genotype was for each cross (since the exact genotypes are not specified). I suggest clarifying the genotypes of the parents and pups and adding a column with expected numbers of offspring of each genotype.

Reviewer #3 (Remarks to the Author):

The authors have addressed my questions adequately in the revised version of their manuscript. I have no further suggestions and recommend this manuscript for publication in Nature Communications.

REVIEWERS' COMMENTS:

Reviewer #2 (Remarks to the Author):

The authors have invested significant effort in addressing the comments of the reviewers and, overall, I am satisfied with the revisions and now support publication. I have a few final requests that I think could improve and/or clarify some of the points made in the manuscript.

We thank the reviewer for providing comments and suggestions with experiments that further tested and strengthened the conclusions in this manuscript.

1. At the end of the section that describes the LPS challenge experiments (line 230), the authors should add an additional sentence that explicitly states that endogenous Firre does not appear to be necessary for the normal inflammatory response to LPS challenge. This is clear from Figure 2f, which shows that WT and Firre KO mice produce equivalent cytokines after LPS administration.

We thank the reviewer for this comment and agree that it is important to point this out in the text and have added the following:

(lines 226-229): “Collectively, these results indicate that endogenous Firre does not appear to be necessary for a physiological inflammatory response to LPS, but that ectopic levels of Firre RNA can modulate this inflammatory response in vivo independent of genomic context, consistent with an RNA-based role for Firre.”

In addition, because the reviewer concluded that WT and *Firre* KO mice produce equivalent cytokines upon exposure to LPS (Figure 2F), we elected to perform an additional replicate experiment to be more confident that no difference was present. We therefore repeated the cytokine analysis in WT (n=5), *Firre* KO (n=5) along with *Firre*^{OE} no dox (n=3) and *Firre*^{OE} dox (n=3) mice injected with LPS. Consistent with the previous results, we find that the cytokine response does not differ in WT and *Firre* KO models. We have now included this additional data, as shown in Figure 2F (below).

In preparing the revised manuscript we re-examined the source data for every panel in the manuscript, and we noted an error in the previous version of Figure 2F. In the earlier version, cytokine measurements from saline injected WT and KO mice were overlaid in the WT LPS and KO LPS columns (these were the two lowest data points in each respective group). We apologize for this error, and have corrected it in the current version of the figure, where the saline control mice are shown in the right-most columns for TNF, IL-12p40, and MIP-2. Importantly, this does not change the interpretation or significance of any of these data, and has no effect on any of the conclusions in the paper.

Figure 2F: (F) Cytokine measurements in serum at 5 hours post intraperitoneal (i.p.) injection of 5 mg/kg LPS (broad-acting) in WT (n=8), Δ Firre (n=8), *Firre*^{OE} control diet (n=6), *Firre*^{OE} dox diet (n=5 to 6), and saline injected WT (n=2) and Δ Firre (n=2) (two independent experiments shown).

2. Also regarding the LPS challenge experiment, it is usually recommended to include an rtTA only + Dox control (to rule out any unexpected effects of Dox treatment). If the authors have such data, it would strengthen their results if they added it to the manuscript. In the future, I recommend including this control.

We did not perform an additional control for rtTA only +dox in the survival experiments presented in Figure 2H. However, we tested the effect of dox in unchallenged *Firre* overexpressing mice (Supplementary Figure 4B) and did not find an increase in cytokines relative to a control group.

3. In the experiments shown in Figure 4d-f (rescue of gene expression changes in CLPs) the authors emphasize that 78 of the 89 genes that were differentially expressed in *Firre* KO CLPs were reciprocally regulated in *Firre*-rescue CLPs. Nevertheless, the comparison of *Firre*-rescue to *Firre* KO revealed 4656 differentially expressed genes, raising concerns that they are not really seeing a specific rescue. 4656 genes probably represents 1/3 to 1/2 of all genes expressed in CLPs. Given this massive change in gene expression, it is less impressive that the small number of genes that are aberrantly expressed in *Firre* KO cells show some reversal of this effect. In the final manuscript, the authors should discuss the possibility that some of the apparent rescue of gene expression that they observed in *Firre*-rescue mice is due to the extremely widespread changes in gene expression observed in those animals rather than specific rescue of the 89 differentially expressed genes detected in *Firre* KO mice.

We thank the reviewer for this comment and agree that while this result was highly statistically significant ($P=2.2e-16$, Fisher exact test), we now discuss the possibility of some of the rescue relating to widespread changes in gene expression. We now discuss this point in the Discussion:

(lines 374-380): *“By using compound genetic approaches, we found that overexpression of Firre from a transgene in the Firre-deficient background was sufficient to rescue physiological and molecular phenotypes in Δ Firre CLPs in vivo. The molecular rescue of genes found differentially expressed in Δ Firre CLPs produced a highly significant result ($P=2.2e-16$, Fisher exact test); however, we note that the widespread changes in gene expression observed in the CLPs from animals only expressing transgenic Firre RNA could also formally contribute to this effect.”*

4. Table 1 is confusing. It is unclear what the expected number of mice of each genotype was for each cross (since the exact genotypes are not specified). I suggest clarifying the genotypes of the parents and pups and adding a column with expected numbers of offspring of each genotype.

We apologize for the confusion for Supplementary Table 1. Because *Firre* is an X chromosome gene we sought to examine whether the proportion of male and female progeny was affected upon deletion of the *Firre* locus and overexpression of the *Firre* RNA. The random nature of the transgenic insertion makes it not possible to predict the exact proportions of expected genotypes in the progeny to assess Mendelian Inheritance. We have now clarified the purpose of Supplementary Table 1 in the text, and indicated the genotype and sex of the parent mice being mated under the “Mating Genotype” column and the genotypes of the progeny in the “Progeny Genotype” column as well as update the figure legend:

(lines 139-143): *“Since Firre is found on the X chromosome, we first sought to determine if deletion of the locus had an effect on the expected sex ratio of the progeny. Matings*

between Δ Firre mice produced viable progeny and had a normal frequency of male and female pups (Supplementary Table 1) that did not exhibit overt morphological, skeletal, or weight defects (Supplementary Fig. 2).”

(lines 162-163): “Moreover, matings between *tg(Firre)* and *rtTA* mice fed a dox diet produce viable progeny that overexpress *Firre* at expected male and female frequencies (Supplementary Table 1).”

Mating Genotype	Diet	Litters	Total pups	Mean litter size (\pm sd)	Progeny Genotype	Total number pups per genotype				P value
						♂	♀	n.d		
♂ Firre ^{+/y} x ♀ Firre ^{+/+}	Normal	6	39	6.5 \pm 1.3	Wildtype	39	20	19	0	0.873
♂ Firre ^{+/y} x ♀ Firre ^{-/-}	Normal	10	68	6.8 \pm 1.9	Δ Firre	68	30	38	0	0.332

♂ rtTA x ♀ Firre ^{OE}	Control	10	66	6.6 \pm 1.5	Firre ^{OE} ; rtTA	21	08	13	0	0.2752
					Firre ^{OE}	23	11	07	05	
					rtTA	11	03	01	07	
					Wildtype	09	02	04	03	
n.d	02									

♂ rtTA x ♀ Firre ^{OE}	Dox.	33	206	5.7 \pm 1.7	Firre ^{OE} ; rtTA	41	19	22	0	0.6394
					Firre ^{OE}	76	35	28	13	
					rtTA	22	11	10	01	
					Wildtype	57	17	11	29	
n.d	10									

Supplementary Table 1. Loss or gain of *Firre* expression does not alter sex distribution. (A) Deletion of the *Firre* locus does not significantly alter the distribution of male and female progeny from matings between male and female Δ *Firre* mice, or in matings between male and female *Firre* WT mice. **(B)** Matings between male *rtTA* and female *Firre*^{OE} mice on a dox or control diet do not show a significant difference in the distribution of male and female progeny that overexpress or do not overexpress *Firre* (*Firre*^{OE}; *rtTA*). Tissue collection for genotyping performed at P7. Litter size shown as mean with standard deviation (s.d.), not determined (n.d.), Chi-square statistic reported (p-value).

With regard to the reviewer’s suggestion, the Δ *Firre* model enables us to test Mendelian inheritance of the alleles. We examined 100 pups from 13 different litters and this shows that Δ *Firre* alleles are inherited at the expected Mendelian distribution which we present below in Reviewer Figure 1.

Reviewer Figure 1: Δ *Firre* mice are viable and come out at expected Mendelian ratios

Mating genotype	Progeny genotype	Observed	Expected
		Number of pups	Number of pups
Firre ^{+/y} x Firre ^{+/-}	Firre ^{+/y}	24	25
	Firre ^{-/y}	21	25
	Firre ^{+/-}	22	25
	Firre ^{-/-}	33	25
Total number of pups		100	100
Total number of litters		13	
Chi-square value		3.6	
P value		0.308	

Reviewer #3 (Remarks to the Author):

The authors have addressed my questions adequately in the revised version of their manuscript. I have no further suggestions and recommend this manuscript for publication in Nature Communications.

We thank the reviewer for providing comments and suggestions with experiments that further tested and strengthened the conclusions in this manuscript.